# Solar system expansion and strong equivalence principle as seen by the NASA MESSENGER mission

Antonio Genova [1,2], Erwan Mazarico [2], Sander Goossens[2,3], Frank G. Lemoine[2], Gregory A. Neumann[2], David E. Smith[1] & Maria T. Zuber[1]

The NASA MESSENGER mission explored the innermost planet of the solar system and obtained a rich data set of range measurements for the determination of Mercury's ephemeris. Here we use these precise data collected over 7 years to estimate parameters related to general relativity and the evolution of the Sun. These results confirm the validity of the strong equivalence principle with a significantly refined uncertainty of the Nordtvedt parameter $\eta = (-6.6 \pm 7.2) \times 10^{-5}$. By assuming a metric theory of gravitation, we retrieved the post-Newtonian parameter $\beta = 1 + (-1.6 \pm 1.8) \times 10^{-5}$ and the Sun's gravitational oblateness, $J_{2\odot} = (2.246 \pm 0.022) \times 10^{-7}$. Finally, we obtain an estimate of the time variation of the Sun gravitational parameter, $G\dot{M}_\odot / GM_\odot = (-6.13 \pm 1.47) \times 10^{-14}$, which is consistent with the expected solar mass loss due to the solar wind and interior processes. This measurement allows us to constrain $|\dot{G}|/G$ to be $<4 \times 10^{-14}$ per year.

[1] Department of Earth, Atmospheric and Planetary Sciences, Massachusetts Institute of Technology, Cambridge, MA 02139, USA. [2] NASA Goddard Space Flight Center, Greenbelt, MD 20771, USA. [3] Center for Research and Exploration in Space Science and Technology, University of Maryland, Baltimore County, Baltimore, MD 21250, USA. Correspondence and requests for materials should be addressed to A.G. (email: antonio.genova@nasa.gov)

Mercury's role in testing theories of gravitation has always been crucial because the strong gravitational mass of the Sun creates notable perturbations on its orbit. The precession of the closest distance of Mercury to the Sun (perihelion) first highlighted the limits of Newtonian physics and later validated the predictions of Einstein's theory of general relativity (GR)[1]. The precession of Mercury's perihelion is primarily due to third-body perturbations from other planets (~531.63" per Julian century[2]), and the relativity effect produces an additional perihelion shift of ~42.98" per Julian century[3]. The relativistic corrections to Mercury's heliocentric acceleration can be formulated based on the parameterized post-Newtonian (PPN) parameters $\beta$ and $\gamma$, which respectively measure the nonlinearity in superposition of gravity and space-time curvature produced by a unit rest mass. Both parameters are zero in the Newtonian formulation and equal to 1 in GR (Methods).

The Sun's interior structure and dynamics also affect Mercury's trajectory. The solar gravitational oblateness $J_{2\odot}$ and the angular momentum $S_\odot$ are responsible for additional precession rates of ~0.029" per Julian century[4] and ~0.002" per Julian century[5], respectively. The latter perturbation, which is known as the gravitomagnetic Einstein–Lense–Thirring (ELT) effect, is related to the distortion of space-time induced by the rotation of the Sun.

In practice, strong correlations between $\gamma$, $\beta$, $J_{2\odot}$, and $S_\odot$ limit the combined estimation of these parameters since they all primarily affect Mercury's orbit through the precession of its perihelion. For this reason, a priori assumptions are necessary to disentangle the effect of each parameter. The Nordtvedt parameter $\eta$, related to the equivalence principle (EP), can be used as a constraint between the PPN parameters $\gamma$ and $\beta$[6]. The relationship between these coefficients is:

$$\eta = 4\beta - \gamma - 3 \qquad (1)$$

if we assume spatial isotropy, which implies that the PPN parameters $\alpha_1$ and $\alpha_2$ are equal to 0.

The orbit of Mercury is well-suited to test the EP, which describes the equality between gravitational and inertial masses. The EP has been partially demonstrated by laboratory experiments, to a precision of ~$1 \times 10^{-13}$ with recent torsion-balance tests[7]. However, these precise results only concern the weak EP, which is based on Galileo's postulate that different objects fall with the same acceleration in a uniform gravitational field, independent of their composition and structure. Einstein extended this concept in his development of GR by introducing the strong EP (SEP). The SEP states that a uniform gravitational field is locally indistinguishable from an accelerated reference frame[8]. The contribution of the SEP to the gravitational-to-inertial mass ratio depends on the self-gravitational energy ($\Omega_B$) and the rest energy of the body ($m^I c^2$), as follows:

$$\frac{m^G}{m^I} = 1 + \eta \frac{\Omega_B}{m^I c^2} \qquad (2)$$

where $m^G$ and $m^I$ are the gravitational and inertial masses, respectively, $\Omega_B$ is proportional to $G(m^G)^2 R^{-1}$, $G$ is the gravitational constant, $c$ is the speed of light, and $R$ is the radius of body B, respectively. The Nordtvedt parameter $\eta$ must be zero to validate the SEP. To prove the SEP, the test mass used in the experiment needs to be sufficiently large so that the self-gravitational force is not negligible. For this reason, tests at the scale of the solar or planetary system are suitable to prove the SEP.

The most accurate estimations of $\eta$ have been retrieved from lunar laser ranging (LLR) over the past 40 years[9–12]. The latest solution validates the SEP with an uncertainty of $\sigma_\eta \sim 3.0 \times 10^{-4}$ (Table 1). The coupling of the gravitational attraction of the Sun on the Earth–Moon system with the self-gravitational force of the Earth would provide a significant perturbation in the case of SEP violation. This effect would be measurable with LLR mm-precision data of the Earth–Moon distance[12].

An equivalent dynamical effect on Mercury's orbit is due to the coupling between the Sun's self-gravitational force and the gravitational attraction of other planets, mainly Jupiter. However, the main effect that a SEP violation has on the ephemeris of Mercury results from the implied redefinition of the solar system barycenter (SSB), which is negligible in the Earth–Moon case (Methods). A Nordtvedt parameter $\eta$ of $1 \times 10^{-5}$ results in discrepancies in the Mercury–Earth relative distance of ~3 m after 2 years[13]. Thus, the knowledge of Mercury's ephemeris to better than 1 m can yield better constraints on possible SEP violations than LLR. Furthermore, this dynamical perturbation of the Nordtvedt parameter is less correlated with other forces and thus separates the effects of $J_{2\odot}$ and $\beta$, given the constraint of Eq. 1.

The study of Mercury's orbit with a long-duration data set also gives a unique opportunity to detect the time variation of the

**Table 1 Current knowledge of general relativity and heliophysics parameters**

| | Recent values | References |
|---|---|---|
| $GM_\odot$ (km$^3$ s$^{-2}$) | $132712440043.754 \pm 0.14$ | Latest solution of the INPOP (Intégration Numérique Planétaire de l'Observatoire de Paris) planetary ephemerides[20]. |
| $J_{2\odot}$ ($\times 10^{-7}$) | $2.30 \pm 0.25$ | Helioseismology result based on the theory of slowly rotating stars[22]. |
| | $2.20 \pm 0.03$ | Helioseismology result with satellite and Earth-based measurements[23]. |
| $S_\odot$ ($\times 10^{39}$ kg m$^2$ s$^{-1}$) | $190 \pm 1.5$ | |
| $\gamma - 1$ | $(2.1 \pm 2.3) \times 10^{-5}$ | Cassini superior solar conjunction experiment[27]. |
| $\beta - 1$ | $-6.7 \pm 6.9) \times 10^{-5}$ | Numerical estimation with INPOP13c[20]. |
| | $(1.2 \pm 1.1) \times 10^{-4}$ | Lunar laser ranging (LLR) experiment[11]. |
| $\eta$ | $(1.0 \pm 3.0) \times 10^{-4}$ | LLR analysis based on refined modeling[12]. |
| $\lvert \dot{G} \rvert / G$ (per year) | $(1.0 \pm 2.5) \times 10^{-13}$ | 21-year timing of the millisecond pulsar J1713+0747[26]. |
| | $(6.0 \pm 11.0) \times 10^{-13}$ | INPOP[20] and ephemerides of the planets and the Moon (EPM2011)[25]. |
| | $< 0.8 \times 10^{-13}$ | |
| $G\dot{M}_\odot / GM_\odot$ (per year) | $(-0.50 \pm 0.29) \times 10^{-13}$ | INPOP[20] |
| | $(-0.63 \pm 0.43) \times 10^{-13}$ | EPM2011[25] |
| $\dot{M}_\odot / M_\odot$ (per year) | $(-1.124 \pm 0.25) \times 10^{-13}$ | Combined estimation of Sun's luminosity and solar wind. |

These quantities were obtained from a variety of dedicated investigations, including helioseismology and LLR experiments. The uncertainties reported in the table are 1-$\sigma$. The $GM_\odot$ and $J_{2\odot}$ adopted in this study as a priori are the JPL DE432 values, $GM_\odot = 132712440041.9394$ km$^3$ s$^{-2}$ and $J_{2\odot} = 2.1890 \times 10^{-7}$, which were reported without formal uncertainties. The $\dot{M}_\odot / M_\odot$ value is given by the mass loss rates induced by Sun's luminosity $\dot{M}_\odot / M_\odot = -0.679 \times 10^{-13}$ per year[28] and solar wind $\dot{M}_\odot / M_\odot = -(0.2 - 0.69) \times 10^{-13}$ per year[28, 29], respectively. The uncertainty is mainly related to the solar wind contribution

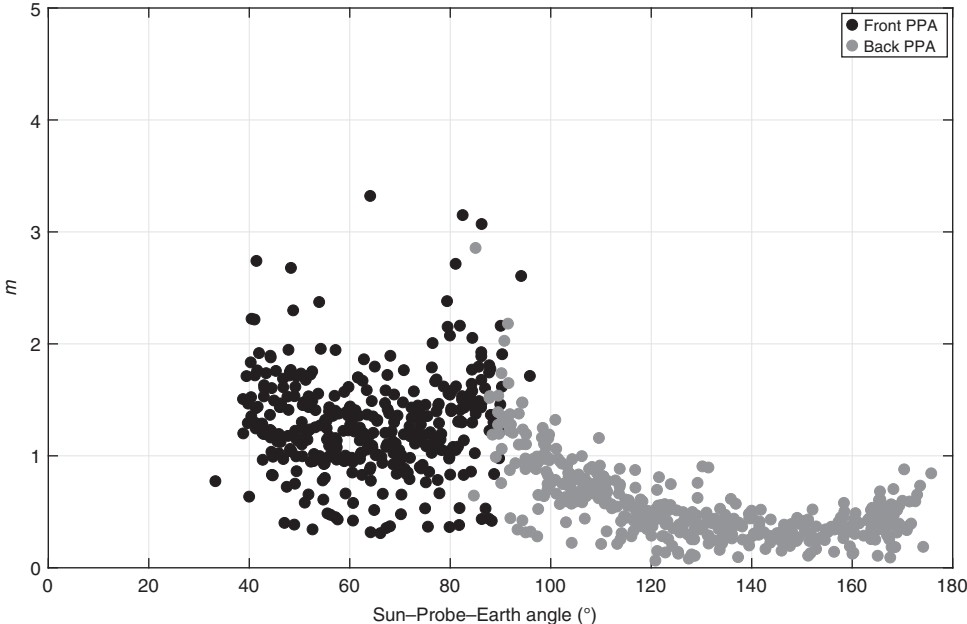

**Fig. 1** Noise level of the MESSENGER range data. RMS of range measurements as a function of the Sun–Probe–Earth angle, which illustrates the effect of the solar plasma on the data noise. Lower SPE angles produce higher noise since the signal passes through dense solar plasma closer to the Sun. The data collected near superior solar conjunction (SPE < 35°) were not included in the analysis. The figure also shows the antennas that were used to provide the downlink to the DSN station. The range data were always collected during tracking passes with fanbeam for uplink and PPAs for downlink reducing thermal noise effects

gravitational constant $G$. The estimation of $\dot{G}/G$ is not strongly correlated with other relativistic and solar parameters because its effect is quadratic in time. However, Mercury's orbit is perturbed by the combined effect of secular changes in $G$ and $M_\odot$ as follows:

$$G\dot{M}_\odot/GM_\odot = \dot{G}/G + \dot{M}_\odot/M_\odot, \qquad (3)$$

where $\dot{M}_\odot/M_\odot$ is the Sun's mass loss due to solar radiance and wind. A perturbation on Mercury's orbit induced by a $G\dot{M}_\odot/GM_\odot$ of $5 \times 10^{-14}$, which is ~10% of the Sun's expected mass loss[14], is on the order of ~2 m after 2 years, when projected on the Earth–Mercury line-of-sight. An estimated time variation of $GM_\odot$ combined with a $\dot{M}_\odot/M_\odot$ value from heliophysics studies improves the knowledge of $\dot{G}/G$. Such a study of heliophysics and relativity requires precise observations of Mercury's position and velocity over an extended period of time.

In this study, we focused on the radio science data of the NASA Mercury Surface, Space Environment, Geochemistry, and Ranging (MESSENGER) mission to investigate the interior structure and evolution of the Sun ($GM_\odot$, $J_{2\odot}$, and $\dot{M}_\odot/M_\odot$) and theory of gravitation ($\beta$, $\eta$, and $\dot{G}/G$). Our results show improved estimates of the solar gravitational oblateness and the mass loss rate that are consistent with helioseismology and heliophysics theoretical studies, respectively. The accurate measurement of the time variation of the solar gravitational parameter enabled us to constrain $|\dot{G}|/G$ to be lower than $4.0 \times 10^{-14}$ per year. Furthermore, we determined the Nordtvedt parameter $\eta$ with a refined uncertainty that demonstrates that there are no violations of the SEP at the level of ~$6$–$7 \times 10^{-5}$.

## Results

**MESSENGER and Mercury-combined orbit determination.** The MESSENGER mission collected spacecraft radio tracking data near Mercury between January 2008 and April 2015, which are well-suited to improve the ephemeris of the planet[15]. These data are range-rate (or Doppler) observables that measure the

relative velocity in the line-of-sight between the spacecraft and a Deep Space Network (DSN) Earth station, and range observables of the relative distance between the spacecraft and the DSN station.

Doppler observables have been used extensively to determine the trajectory of spacecraft for navigation and geophysical parameter estimation, e.g., the gravitational field of Mercury[16]. On the other hand, range observables bear on the knowledge of Mercury's orbit. Range measurements have been analyzed by the Solar System Dynamics Group of the Jet Propulsion Laboratory (JPL), Institute of Applied Astronomy of the Russian Academy of Science, and the Institut de mécanique céleste et de calcul des éphémérides to determine the ephemeris of Mercury, estimating relativistic and heliophysics perturbative forces[17–20].

Parallel and independent investigations so far have been conducted to exclusively determine either Mercury's geophysics or its ephemeris. The estimation of Mercury's gravity field relied on the assumption of planet's orbits and GMs, including Mercury, from JPL development ephemeris (DE). On the other hand, the ephemeris work processed spacecraft range measurements only by using a pre-converged MESSENGER trajectory. Although both methods have successfully been used for interplanetary orbit determination, their piecemeal combination is not the best approach in the case of Mercury. Systematic errors in the MESSENGER orbits directly enter the range data, and are then absorbed into Mercury's estimated position, since the spacecraft trajectory is not adjusted in the ephemeris solution. Conversely, a mismodeling of Mercury's ephemeris leads to imperfect geophysical solutions.

We numerically integrate the spacecraft and planet orbits simultaneously in order to provide a comprehensive solution that includes geophysical, heliophysics, and relativity results together with their associated covariances. Here we focus on the results that provide new information on the interior of the Sun and on gravitational theories.

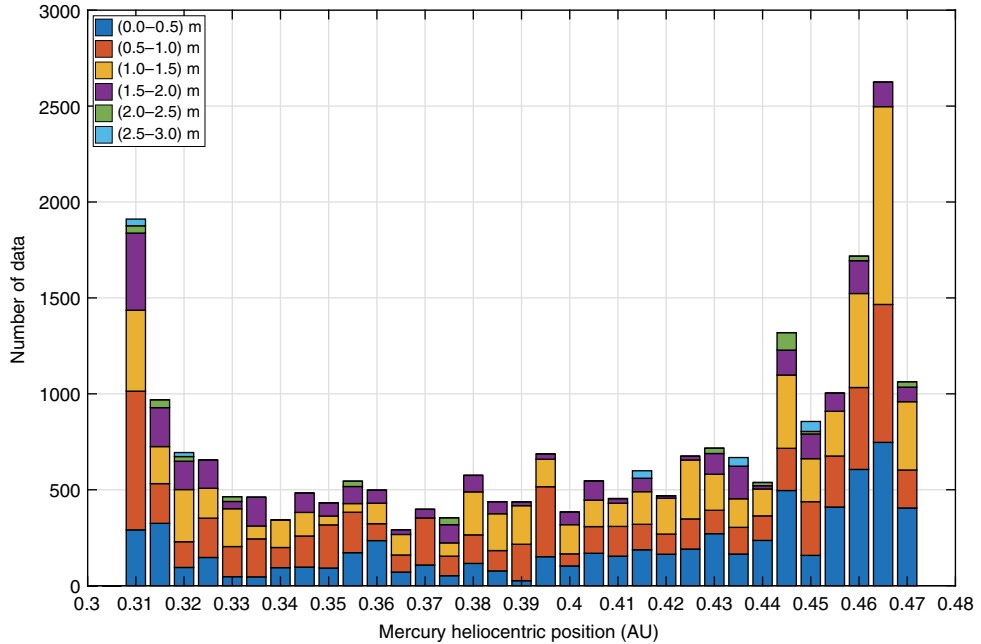

**Fig. 2** Data distribution throughout Mercury's orbit. Number of the analyzed measurements as function of the Mercury distance from the Sun in AU. Colors indicate the noise level distribution during each phase bin of Mercury's orbit. The greater part of the data was collected close to Mercury's perihelion and aphelion

The accuracy of the heliophysics and relativity results largely depends on the precision of the range data. Measurement noise is incurred by the electronics of the radio frequency (RF) telecommunications subsystem onboard the spacecraft that relays spacecraft telemetry and performs as a radio science instrument[21]. The MESSENGER RF subsystem operated at X-band frequencies (7.2 GHz uplink, 8.4 GHz downlink) and its two opposite-viewing phased-array antennas (PAAs) were used to conduct the range data campaigns (Methods).

Figure 1 shows the level of noise of the range data collected over the entire MESSENGER mission. Each point represents the range data RMS (root mean square) during each full tracking pass, which usually provides one measurement every 5 min for several hours, as a function of the Sun–Probe–Earth (SPE) angle. The relative position of the Sun and Earth during MESSENGER radio observations strongly controls the tracking data quality. The solar plasma causes phase scintillations in the RF signal, increasing the noise of both range-rate and range measurements. At low SPE angles (near superior solar conjunctions), the Sun is located between Mercury and the Earth, and the mean level of noise increases from <0.5 m for 90° < SPE < 180° to ~1.5 m for 35° < SPE < 90° (Fig. 1). MESSENGER orbits close to superior solar conjunctions (SPE < 35°) are not included in the solution because of plasma-induced range errors higher than 3 m. The exclusion of these data does not degrade the phase sampling of Mercury's orbit over the full MESSENGER mission. Figure 2 shows a histogram of the number of the processed measurements vs. the Mercury orbital phase. A great portion of data was collected in proximity to Mercury's perihelion and aphelion, enabling precise measurements of the precession induced by solar and relativistic effects. The rest of Mercury's orbit was evenly sampled, and the level of noise is sufficiently uniform for all orbital phases.

This MESSENGER data set is used to determine at the same time the solar and relativistic parameters that provide fundamental information on the interior structure of the Sun ($GM_\odot$, $J_{2\odot}$, $S_\odot$, and $\dot{M}_\odot/M_\odot$) and theory of gravitation ($\gamma$, $\beta$, $\eta$,

and $\dot{G}/G$). Multiple separate experiments have previously established a comprehensive survey of these fundamental physics effects. Table 1 summarizes recent estimates of these parameters. Helioseismology studies[22,23] enabled precise measurements of $S_\odot$ and $J_{2\odot}$, which have been also recovered with ephemerides analysis[17–20]. Planetary ephemeris investigations, furthermore, provided the best estimates for $\beta$[20,24,25] and $GM_\odot/GM_\odot$[20,25]. Several of these studies do not include the ELT effect, so their estimated $J_{2\odot}$ must be scaled. A more recent determination of Mercury's ephemeris reported the estimation of $J_{2\odot}$ by accounting for ELT accelerations[24]. LLR provided accurate estimates of $\eta$ and $\dot{G}/G$[9–12], which has also been determined by astrophysical studies[26]. The Cassini mission achieved the most precise measurement of the PPN parameter $\gamma$ through the analysis of radio tracking data near superior solar conjunction[27]. Although the MESSENGER data are not strongly sensitive to $\gamma$ and the Sun's angular momentum, $S_\odot$, this investigation provides a unique opportunity to simultaneously improve the knowledge of $GM_\odot$, $J_{2\odot}$, $\beta$, $\eta$, and $\dot{M}_\odot/M_\odot$.

**Heliophysics and relativity solutions**. The main effect of $\gamma$ on the radio tracking data is the deflection and delay of photons by the curvature of space time produced by the Sun, and it is best measured at SPE angles lower than 10°. However, this geometry yields high plasma noise so the estimation of the PPN parameter $\gamma$ with MESSENGER data is not feasible. Given the $\gamma$-effect on Mercury's dynamical equations cannot be separated from the dynamical perturbation due to the PPN parameter $\beta$ nor ignored, $\gamma$ was thus fixed to 1, while still considering the uncertainty obtained by Cassini[27].

The strong correlation between Mercury's orbital perturbations due to the Sun's gravitational oblateness and the ELT effect also does not allow the determination of the angular momentum of the Sun with the MESSENGER radio tracking data. The a priori value of $S_\odot$ adopted in this study is $190 \times 10^{39}$ kg m² s⁻¹. A covariance analysis shows that the MESSENGER data sensitivity

**Table 2 A priori and estimated values, and uncertainties from the global estimation of the GR and heliophysics parameters**

| | A priori values | Estimated values | Formal uncertainties | Sensitivity to change of planetary ephemerides |
|---|---|---|---|---|
| $GM_\odot$ (km³ s⁻²) | 132712440041.9394 | 132712440042.2565 | 0.35 | 0.87 |
| $J_{2\odot}$ (×10⁻⁷) | 2.1890 | 2.246 | 0.02 | 0.02 |
| $\beta-1$ (×10⁻⁵) | 0 | −1.625 | 1.8 | 1.57 |
| $\eta$ (×10⁻⁵) | 0 | −6.646 | 7.2 | 6.24 |
| $\dot{G}M_\odot/GM_\odot$ (×10⁻¹⁴ per year) | 0 | −6.130 | 1.47 | 3.14 |

The formal uncertainties are given by the covariance matrix of the least-square solution, which does not include possible mismodeling of GMs and states of the other planets, and asteroids of the solar system. The third column reports the maximum discrepancies between solutions that we obtained by using the JPL DE430, DE432, or DE436 ephemerides to model the third-body perturbation of the planets. The ephemerides of the asteroids are based on the JPL AST343DE430[17] for the three cases

to the ELT effect yields $\sigma_{S_\odot} \sim 40 \times 10^{39}$ kg m² s⁻¹, which is ~30 times larger than the current best knowledge (Table 1). For this reason, the angular momentum of the Sun is not adjusted, but the ELT effect is of course included in the integration of both the Mercury and MESSENGER orbits. The ELT effect was not modeled in the JPL DE432 ephemerides[18].

Our results are based on a global combined estimation of MESSENGER- and Mercury-related orbital dynamics (Methods). Table 2 shows the a priori and estimated values and uncertainties of the heliophysics and relativistic parameters. The Sun's $GM$ and $J_2$ estimates are in good agreement with previous works based on Mercury's ephemeris analysis[20,24,25]. The solar gravitational flattening is notably improved and consistent with helioseismology results, which were based on solar internal rotation measurements[22,23]. By applying Eq. 1 as constraint, we assume a metric theory of gravitation. The Nordtvedt relation enables a highly accurate recovery of $J_{2\odot}$ and $\beta$ leading to a formal uncertainty of the gravitational flattening refined by, at least a factor of 3 compared to previous ephemeris studies[20,24,25]. However, the correlation between $J_{2\odot}$ and $\beta$ is still high (~0.9, Supplementary Table 1) because the estimation of $\eta$ is limited by the accuracy of the range data (Methods). Four different cases were studied to assess the effects of a priori knowledge or constraints, if we do not assume a metric theory of gravitation or if we assume that $\beta-1$, $\eta$, or both parameters are equal to 0. These tests generalize our results further and are shown in Supplementary Table 2. The Nordtvedt equation significantly benefits the estimation of $\beta$ and $J_{2\odot}$, but the $\eta$ and $GM_\odot/GM_\odot$ estimates are always stable and near the values shown in Table 2. We note that an unconstrained solution yields a near-unity $\beta$-$J_{2\odot}$ correlation and values of $\beta-1 = (-1.43 \pm 1.47) \times 10^{-4}$ and $J_{2\odot} = (2.10 \pm 0.15) \times 10^{-7}$. In case we do not adjust for $\beta$ and $\eta$, the Sun's gravitational oblateness converges to $(2.271 \pm 0.003) \times 10^{-7}$ that is still within ~1-$\sigma$ of the constrained solution.

Both constrained and unconstrained solutions are consistent with Einstein's theory of GR. GR predictions of $\beta$ and $\eta$ values are within 1-$\sigma$, as reported in Table 2. These results enable substantial enhancement of both $\beta-1$ and $\eta$ estimates, which are ~7 and ~5 times closer to 0 than LLR solutions, respectively. The knowledge of the PPN parameter $\beta$ in this study is now comparable to the Cassini solution of the PPN parameter, $\gamma$.

Furthermore, Table 2 reports the estimation of $\dot{G}M_\odot/GM_\odot$ that combines the temporal variations of both $G$ and $\dot{M}_\odot$. The retrieved negative rate is close to the theoretical computations of the Sun's mass loss due to interior processes and solar wind. The fusion cycle that generates energy into the Sun relies on the conversion of hydrogen into helium, which is responsible for a solar mass reduction with a rate of $\sim -0.679 \times 10^{-13}$ per year[28]. On the other hand, the solar wind contribution is more uncertain. The solar cycle significantly influences the solar mass loss rate due to solar wind. Estimates of the mass carried away with the solar wind showed rates between $-(2-3) \times 10^{-14} M_\odot$ per year[28], whereas numerical simulations of coupled corona and solar wind models provided rates between $-(4.2-6.9) \times 10^{-14} M_\odot$ per year[29]. Therefore, a mean value of the total solar mass loss of $-(0.9-1.1) \times 10^{-13} M_\odot$ per year would be expected, since the MESSENGER mission operated during ~2/3 of an entire solar cycle whose maximum occurred in proximity of the end of the mission.

**Discussion**

The estimated $\dot{G}M_\odot/GM_\odot$ represents one of the first experimental observations of the solar mass loss. Previous studies have demonstrated the feasibility of estimating this parameter by adjusting the planetary ephemerides[20,25,28]. Their results, which are consistent with our estimates, were limited by the data availability and possible mismodeling of spacecraft orbits. Our processing of the entire MESSENGER mission increased the solution sensitivity to a variation of $\dot{G}M_\odot/GM_\odot$, which has a quadratic dependence in time. Furthermore, our new technique, which consists of a double-integration and a combined estimation of both planet and spacecraft orbits, mitigates the systematic errors related to the spacecraft position and velocity.

The discrepancy between our solution and the computed $\dot{M}_\odot/M_\odot$ may be interpreted as an indirect measurement of the universal constant time variation. The reconstructed $\dot{G}/G$ is lower than $4.0 \times 10^{-14}$ per year with an uncertainty that is mainly limited by the knowledge of the solar interior evolution ($2\sigma = 5.0 \times 10^{-14}$ per year). This result strengthens the hypothesis that $\dot{G}/G$ is close to 0, improving the estimates of LLR studies by almost an order of magnitude.

To validate the accuracy of these results, we reintegrated the orbit of Mercury with our adjusted values. Figure 3 shows the required corrections to the MESSENGER range data to fit at the noise level shown in Fig. 1. The red, blue, and green dots are the measurement biases needed with the JPL DE430 and DE432 ephemerides, and our solution, respectively. This plot shows major improvements compared to previous JPL ephemerides. The DE430 Mercury trajectory is affected by 80-m amplitude errors that were corrected in the later DE432, with remaining 5–10 m errors. Our reintegrated ephemeris for Mercury, which is available on the NASA Goddard Space Flight Center (GSFC) Planetary Geodynamics Data Archive[30], shows only 0.5–3 m biases over the full mission.

The stability of Mercury's orbit integration also depends on the ephemerides of the other bodies of the solar system that are provided by the JPL ephemerides. Therefore, we evaluated the changes in recovery of the heliophysics and relativity parameters when using DE430 or DE436 for planetary ephemerides and initial state of Mercury (instead of DE432 previously) and modeling the ephemerides of the asteroids with the JPL

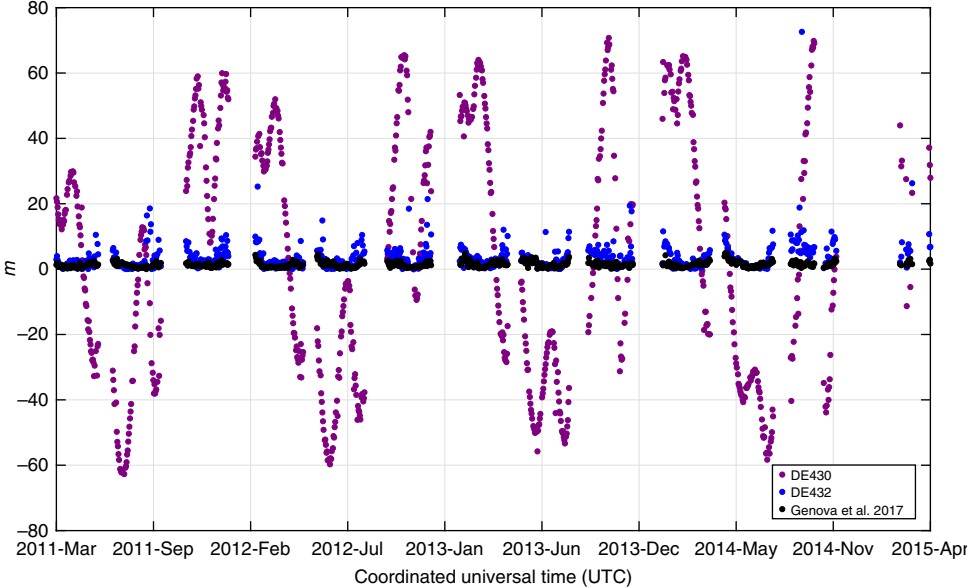

**Fig. 3** Temporal distribution of the range biases with three Mercury's ephemeris. The measurement biases are required to fit the MESSENGER range data at the noise level with the JPL DE430 (purple) and D432 (blue) ephemerides, and our integrated trajectory for Mercury (black). These biases were used to determine the quality of the ephemeris results. After convergence of the global solution, all the adjusted parameters (Methods) are applied in a final iteration, in which the range biases are adjusted instead of the Mercury's initial state, $GM_\odot$, $J_{2\odot}$, $\beta$, $\eta$, and $\dot{GM}_\odot/GM_\odot$. Large range biases suggest significant errors in the planet's ephemeris

AST343DE430[17]. These solutions, which are reported in the Supplementary Table 3, show consistent results with the estimated values in Table 2. The formal uncertainties do not take into account probable errors in the planets' $GM$s and trajectories, as they rely only on the measurement accuracies and the correlation among the adjusted parameters. The fourth column of Table 2 shows the maximum estimation differences between the three cases, which rely on the JPL DE430, DE432, or DE436 for the other planet trajectories. These values may be interpreted as conservative uncertainty bounds that account for pessimistic errors due to mismodeling of $GM$s and orbits of the solar system bodies. The discrepancies between $GM_\odot/GM_\odot$ solutions are slightly larger than the formal uncertainties because of variations in Earth's orbit, Jupiter's $GM$ and orbit, and SSB location between the different JPL DEs that our methodology cannot mitigate. Furthermore, the value of the solar gravitational constant is significantly affected by these discrepancies leading to estimates that are, however, still within 2- and 3-$\sigma$s (Supplementary Table 3).

In conclusion, our analysis of Mercury's ephemeris with the MESSENGER data enhances the knowledge of the relativistic parameter $\eta$, confirming predictions of Einstein's theory of GR. We provide one of the first observations of the solar system expansion due to the solar mass loss. The negative rate of $\dot{GM}_\odot/GM_\odot$ is very close to theoretical computations of the Sun's mass loss rate leading us to significantly constrain the universal constant time variation. These results are mainly limited by the uncertainty in planet and asteroid ephemerides that perturb Mercury's orbit. We demonstrate the potential of measuring the planets' relative distances over decadal timescales to provide a better understanding of the solar system and Sun evolution. To pursue these challenging scientific goals, future investigations employing precision ranging from a dedicated multi-spacecraft constellation at interplanetary scale may provide a leap in planetary science, heliophysics, and theoretical physics[31].

## Methods

**Parametrized post-Newtonian formulation**. The orbital dynamics of planets, satellites, and asteroids relies mainly on the gravitational attraction of the other bodies that are modeled as external point masses. Nevertheless, Newton's law of universal gravitation needs to be modified to include Einstein's relativistic corrections by means of the PPN $n$-body formalism[17]. The acceleration of Mercury due to the interaction with other point masses is, therefore, given by:

$$
\begin{aligned}
\mathbf{a}_M^{PPN} = \sum_{A \neq M} &\frac{\mu_A}{r_{MA}^3}(\mathbf{r}_A - \mathbf{r}_M)\Big\{1 \\
&- \frac{2(\beta+\gamma)}{c^2}\sum_{B \neq M}\frac{\mu_B}{r_{MB}} - \frac{2\beta-1}{c^2}\sum_{B \neq A}\frac{\mu_B}{r_{MB}} \\
&+ \gamma\Big(\frac{v_M}{c}\Big)^2 + (1+\gamma)\Big(\frac{v_A}{c}\Big)^2 - \frac{2(1+\gamma)}{c^2}\mathbf{v}_M \times \mathbf{v}_A \\
&- \frac{3}{2c^2}\Big[\frac{(\mathbf{r}_A - \mathbf{r}_M) \times \mathbf{v}_A}{r_{MA}}\Big]^2 + \frac{1}{2c^2}(\mathbf{r}_A - \mathbf{r}_M) \times \mathbf{a}_A\Big\} \\
&+ \frac{1}{2c^2}\sum_{A \neq M}\frac{\mu_A}{r_{MA}^3}\{[\mathbf{r}_A - \mathbf{r}_M] \times [(2+2\gamma)\mathbf{v}_M \\
&- (1+2\gamma)\mathbf{v}_A]\}(\mathbf{v}_M - \mathbf{v}_A) + \frac{(3+4\gamma)}{2c^2}\sum_{A \neq M}\frac{\mu_A}{r_{MA}}\mathbf{a}_A
\end{aligned}
\tag{4}
$$

where $\mathbf{r}_i$, $\mathbf{v}_i$, $\mathbf{a}_i$, and $\mu_i$ are the position, velocity, and acceleration vectors with respect to the SSB, and the gravitational parameter of the body $i$, respectively, $c$ is the speed of light, and $\beta$ and $\gamma$ are the PPN parameters that measure the non-linearity in superposition of gravity and space-time curvature produced by unit rest mass, respectively. This formulation has been applied to the orbital dynamics of both Mercury and MESSENGER. The bodies included in these integrations are all major bodies, including the Sun, Moon, planets, Pluto, and 343 asteroids in the main belt between Mars and Jupiter. The positions, velocities, accelerations, and gravitational parameters are obtained from the JPL DE432 ephemerides[18] for planets, and the JPL AST343DE430 for asteroids[17]. However, to test the stability of the solution, the global estimation was repeated by using JPL DE430 and DE436 ephemerides as a priori. The results of Mercury's ephemeris, $GM_\odot$, $J_{2\odot}$, $\beta$, $\eta$, and $\dot{GM}_\odot/GM_\odot$ are within the corrected $\sigma$ reported in Table 2.

**Lense–Thirring precession**. The dynamical orbital equations of both Mercury and MESSENGER account for the Lense–Thirring effect due to the Sun's gravitomagnetic field that leads to a secular precession of the heliocentric longitude of the ascending node and argument of pericenter.

This precession is a prediction of GR, and, for this reason, it has been recently renamed as ELT effect. Einstein postulated the frame-dragging in the context of the general theory of relativity stating that non-static stationary distributions of mass-energy affect space-time. In 1918, Josef Lense and Hans Thirring derived the first frame-dragging effect predicting that the rotation of a massive body induces a distortion of space-time. The ELT effect has been measured with the LAGEOS satellites in orbit around the Earth[32], and the gyroscopes of the Gravity Probe B[33]. A test on ELT effects was initially proposed with the NASA mission Juno in orbit about Jupiter[34]. The large angular momentum of the planet induces a significant ELT acceleration; however, it is also highly correlated with perturbations due to Jupiter's orientation, which is currently not sufficiently defined[35].

The ELT effect on Mercury due to the solar gravitomagnetic field is not negligible, and it may theoretically be used to measure the angular momentum of the Sun. Mercury's acceleration due to the ELT effect is:

$$\mathbf{a}_{\mathrm{M}}^{\mathrm{ELT}} = \frac{S_\odot G(1+\gamma)}{r_{\mathrm{M}\odot} c^2} \left[ \frac{3}{r_{\mathrm{M}\odot}^2} \left(\mathbf{r}_{\mathrm{M}\odot} \times \mathbf{v}_{\mathrm{M}\odot}\right)\left(\mathbf{r}_{\mathrm{M}\odot} \times \widehat{\mathbf{p}_\odot}\right) + \left(\mathbf{r}_{\mathrm{M}\odot} \times \widehat{\mathbf{p}_\odot}\right) \right] \quad (5)$$

where $S_\odot$ is the angular momentum of the Sun, $G$ is the universal gravitational constant, $\mathbf{r}_{\mathrm{M}\odot}$ and $\mathbf{v}_{\mathrm{M}\odot}$ are the heliocentric position and velocity vectors of Mercury, respectively, and $\widehat{\mathbf{p}_\odot}$ is the unit vector of the Sun's pole direction, which relies on the right ascension $\alpha_\odot = 286.13°$ and declination $\delta_\odot = 63.87°$ of the pole defined in the International Celestial Reference Frame (ICRF)[36].

The ELT effect on Mercury's orbit is mainly in the radial direction with a maximum acceleration of $\sim 2 \times 10^{-7}$ m s$^{-2}$, assuming $S_\odot = 190 \times 10^{-39}$ kg m$^2$ s$^{-1}$. However, the perturbation induced by the ELT precession is strongly anti-correlated with the effect due to $J_{2\odot}$, and the recovery of $S_\odot$ is unachievable with the estimation of Mercury's ephemeris.

**Strong equivalence principle.** Milani et al.[37] formulated, for the first time, a redefinition of the SSB due to violations of the SEP. This effect causes a significant indirect perturbation on Mercury's orbit that enables an accurate measurement of $\eta$ by adjusting the planet's ephemeris. These results provoked a scientific debate on the consequences of SEP violations for the modeling of planetary ephemerides. Ashby et al.[38] presented an alternative approach that does not fully include the indirect perturbation presented by Milani et al.[37], limiting the contribution of $\eta$ on planet's orbital dynamics. However, current planetary ephemerides studies are based on the hypothesis that the gravitational and inertial masses are equal to compute the SSB location[17].

The SSB represents the origin of the ephemerides reference frame. The assumptions to compute its position are the conservation of mass/energy and the momentum of the solar system. The SSB is, then, approximated as follows[13]:

$$\mathcal{R} = \frac{\sum_j \mu_j^* r_j}{\sum_j \mu_j^*} \quad (6)$$

where $\mathcal{R}$ is equal to 0 if the SSB is the origin of the reference frame, $r_j$ is the relative distance of body $j$ with respect to the SSB, and:

$$\mu_j^* = GM_j^{\mathrm{G}} \left\{ 1 + \frac{1}{2c^2} v_j^2 - \frac{1}{2c^2} \sum_{k \neq j} \frac{GM_k^{\mathrm{G}}}{r_{jk}} \right\} \quad (7)$$

where $GM_j^{\mathrm{G}}$ is the gravitational mass parameter of body $j$, $r_{jk}$ is $|r_k - r_j|$ and $v_j$ is the magnitude of the velocity of body $j$. This formulation is valid only when the SEP is not violated ($\eta = 0$). The inertial masses should be used in the computation of the SSB, as follows, if we neglect terms of order $1/c^2$:

$$\frac{\sum_j M_j^{\mathrm{I}} \mathbf{r}_j}{\sum_j M_j^{\mathrm{I}}} = 0 \quad (8)$$

where $\mathbf{r}_j$ is the position vector of body $j$ with respect to the SSB, and $M_j^{\mathrm{I}}$ is the inertial mass of body $j$. However, these inertial masses are unknown since the masses of the Sun, planets, and satellites are retrieved in space by means of their gravitational pull. A violation of the SEP may lead to an intrinsic mismodeling of the SSB position. To account for this effect, the position of the Sun should be redefined by:

$$\mathbf{r}_\odot = -\frac{1}{\mu_\odot \left(1 - \eta \frac{\Omega_\odot}{M_\odot c^2}\right)} \sum_{j \neq \odot} \left(1 - \eta \frac{\Omega_j}{M_j c^2}\right) \mu_j \mathbf{r}_j \quad (9)$$

where the sum includes planets and asteroids, $\Omega_j$ and $M_j$ are the self-gravitational energy and the mass of body $j$, respectively, and the symbol $\odot$ stands for the Sun. The self-gravitational energy of the Sun, Earth, and Moon are $-3.52 \times 10^{-6}$, $-4.64 \times 10^{-10}$, and $-1.88 \times 10^{-11}$, respectively[39,40]. The self-gravitational energy of the other planets is computed by assuming uniform density $\left(\Omega_j = \frac{3}{5}\frac{GM_j^2}{R}\right)$. We also tested other self-gravitational energy modeling for the other planets, but the $\eta$ estimates only changed within 1-$\sigma$ since the Sun's self-gravitational energy represents the dominant term.

The Sun's position correction (Eq. 9) entails an indirect term in the heliocentric acceleration of Mercury. The partial derivative of Mercury's heliocentric acceleration ($\mathbf{a}_{\mathrm{M}}$) with respect to $\eta$, which enables the estimation of this parameter by adjusting the planet ephemeris, is:

$$\frac{\partial \mathbf{a}_{\mathrm{M}}}{\partial \eta} \cong \sum_{j \neq \mathrm{M}} \mu_j \left(\frac{\Omega_{\mathrm{M}}}{M_{\mathrm{M}} c^2}\right) \frac{\mathbf{r}_{\mathrm{M}j}}{r_{\mathrm{M}j}^3}$$
$$+ \sum_{j \neq \odot} \mu_j \left(\frac{\Omega_\odot}{M_\odot c^2}\right) \frac{\mathbf{r}_{\odot j}}{r_{\odot j}^3}$$
$$+ \sum_{j \neq \odot} \mu_j \left(\frac{\Omega_j}{M_j c^2} - \frac{\Omega_\odot}{M_\odot c^2}\right) \frac{\partial \frac{\mathbf{r}_{\mathrm{M}\odot}}{r_{\mathrm{M}\odot}^3}}{\partial \mathbf{r}_\odot} \mathbf{r}_j \quad (10)$$

where the symbol M stands for Mercury, $r_{kj}$ is the position vector of the relative distance between bodies $k$ and $j$, $\mathbf{r}_\odot$ is the position vector of the Sun with respect to the SSB, and the last term is the indirect effect due to the correction of the SSB position neglecting terms of the order $1/\eta^2$. SEP violations would provide significant perturbations on Mercury's orbit that enable the measurement of $\eta$ to high accuracy, and decorrelate the PPN parameters and $J_{2\odot}$ if the Nordtvedt equation (Eq. 1) is applied as a priori constraint[13]. However, the correlation between $J_{2\odot}$ and $\beta$ is still $\sim 0.9$ even applying the Nordtvedt equation (Supplementary Table 1).

This a priori constraint approach was proposed for the first time in the simulations of the relativity experiment that will be conducted by the European Space Agency (ESA) mission BepiColombo[13]. One year of operations in orbit about Mercury will allow BepiColombo to collect 30-cm precision range data for the determination of Mercury's ephemeris. The results of those simulations showed lower correlation between $J_{2\odot}$ and $\beta$ ($\sim -0.3$) by using the Nordtvedt equation[37]. The stronger effect of this constraint on the BepiColombo solutions is mainly due to the more precise range data that will enable to determine a more accurate estimate of SEP violations. The accuracy of $\eta$ estimation affects directly the correlation between $J_{2\odot}$ and $\beta$, if the constraint is applied. If we assume the Nordtvedt equation and to know $\eta$ at the same level of BepiColombo results ($\sim 10^{-6}$), the correlation between $J_{2\odot}$ and $\beta$ drops to $\sim 0.3$ that is consistent with the simulation of the future ESA mission to Mercury[41].

**Time-variable gravitational constant.** The time-varying gravitational parameter $G\dot{M}_\odot/GM_\odot$ is defined by the sum of the time-variations of the gravitational universal constant $\dot{G}/G$ and the mass of the Sun $\dot{M}_\odot/M_\odot$. The additional term of Mercury's heliocentric acceleration due to $G\dot{M}_\odot/GM_\odot$ is:

$$\mathbf{a}_{\mathrm{M}}^{G\dot{M}_\odot} \cong GM_\odot \left(\frac{G\dot{M}_\odot}{GM_\odot} \Delta t\right) \frac{\mathbf{r}_{\mathrm{M}\odot}}{r_{\mathrm{M}\odot}^3} \quad (11)$$

where $\Delta t$ is the difference between the current epoch and the reference epoch J2000 (1 January 2000 at 1200 UTC), and $\mathbf{r}_{M\odot}$ is the relative position vector between Mercury and the Sun.

**Ephemeris and orbit determination.** The results presented in this paper were obtained with the NASA GSFC orbit determination software GEODYN II, which has been used to determine geophysical parameters of, for example, the Earth, Moon, and Mars. We used GEODYN II to recover previous solutions of Mercury's gravity field, orientation, and tides assuming the JPL DE430 ephemeris of Mercury[17]. To estimate Mercury's ephemeris and the associated heliophysics and relativity parameters, we modified GEODYN II to numerically integrate the orbits of both MESSENGER and the planet Mercury simultaneously.

This software is based on a batch least-squares scheme that allows the combination of all observations within one batch (arc) for the estimation of the parameters of interest. The least-squares technique relies on an adjustment of model parameters to minimize the discrepancies between the computed observables and actual measurements (residuals). If the trajectory of the spacecraft alone is integrated, the only parameters that can be estimated are related to MESSENGER's dynamics around Mercury (e.g., the gravity field of the planet). The simultaneous numerical integration of the planet ephemeris allows the adjustment of other model parameters, such as those from heliophysics and relativity that perturb the orbit of Mercury.

The MESSENGER orbital mission (2011–2015) was partitioned in 1499 1-day arcs. Three additional ~10-day arcs cover the three Mercury flybys in 2008–2009. The range data were weighted according to the contribution of the solar plasma that varied through the mission as expressed by the Sun–Earth–Probe angle. For each arc, the Mercury's ephemeris is continuously integrated from the Flyby 1 initial epoch (7 January 2008 at 0000 UTC). We generated partial derivatives of the following MESSENGER-related parameters: spacecraft initial states, areas of the spacecraft sunshield, and solar panels, Mercury's gravity field up to degree and order 100 in spherical harmonics, and Mercury's Love number $k_2$ and orientation (pole's right ascension and declination). We also computed partial derivatives of the following Mercury-related parameters: planet's initial state, $GM_\odot$, $G\dot{M}_\odot/GM_\odot$, $J_{2\odot}$ and $S_\odot$, PPN parameters $\beta$ and $\gamma$, and Nordtvedt's parameter $\eta$.

The individual normal equations of all these arcs were combined and inverted to yield the final estimates of the geophysical, heliophysics, and relativity parameters. The orbit of the Earth is not integrated and adjusted in this study since the orbital accuracy of the Earth from the JPL DE432 ephemerides is comparable to the precision of the MESSENGER data.

**MESSENGER data set.** The data processed in this paper include the three MES-SENGER flybys around Mercury, and the whole orbital mission. The three flybys occurred on 14 January 2008, 6 October 2008, and 29 September 2009, respectively. MESSENGER was inserted in a highly eccentric and near-polar orbit about Mercury on 18 March 2011. The initial period was ~12 h and the orbital periapsis was at ~200-km altitude and ~60°N latitude. Orbit-correction maneuvers (OCMs) were required to maintain the periapsis between 200 and 500 km for the first year of operations. The third-body perturbation of the Sun combined with the high

eccentricity of the orbit led to a significant drift of the periapsis altitude and latitude.

The mission was extended for a second year in March 2012. The OCMs became less frequent, and one of them was used to reduce the orbital period to ~8 h. A second extended mission (XM2) started in March 2013 and included a low-altitude campaign until Mercury impact on 28 April 2015. The fuel reserves enabled the spacecraft to maintain periapsis altitudes as low as 15–25 km for several weeks. NASA's DSN stations tracked the spacecraft during part of these passages from April to October 2014 leading to accurate measurements of Mercury's gravity at altitudes between 25 and 100 km. In the last 6 months of the mission, the closest approaches of MESSENGER were occulted by Mercury and were thus not visible from the Earth. However, additional range-rate and range measurements were collected at low altitudes between 75 and 100 km.

The data included in this study were collected over ~900 days. The greater part of the excluded data is because of high levels of plasma noise in proximity of superior solar conjunctions (SPE < 35°). Other arcs were also omitted because of the presence of OCMs or reaction wheel momentum desaturation maneuvers that imparted significant ΔVs leading to significant orbital errors.

**Range-rate and range measurements**. The analysis of the range data to estimate Mercury's ephemeris relies strongly on the accuracy of MESSENGER orbital reconstruction. The data collected during XM2, especially, are very sensitive to the quality of the spacecraft orbits. Uncompensated gravity anomalies of Mercury's gravity field may affect significantly the range residuals leading to inaccurate ephemeris solutions. To mitigate the effects of MESSENGER orbital errors in the determination of Mercury's ephemeris, both range-rate and range data have been analyzed in this study. This data set includes two-way and three-way coherent range-rate and two-way coherent range measurements. The difference between two- and three-way data is only related to the receiving station. The signal is transmitted by the DSN station and sent coherently back to the same (two-way) or a different (three-way) station by the spacecraft deep space transponder (DST). The two-way configuration guarantees highly accurate data thanks to the H-masers at the DSN ground stations. The three-way data require additional bias corrections due to the time delay between the oscillators at the transmitting and receiving stations. The biases of the three-way range-rate data are adjusted in the solution to mitigate this error source.

The Earth-spacecraft radio link was supported by diametrically opposite-facing PAAs for the high-gain downlink signal, and two fanbeam antennas to provide medium-gain uplink and downlink. Four low-gain antennas were also used to enable the instrumentation pointing towards the planet surface during tracking periods. However, the range data campaigns were always conducted with the front and back PPAs, as shown in Fig. 1. The gain level of the antennas influences significantly the level of noise of the range-rate data[21]. A major source of error for the range data is the internal spacecraft delay that was measured during ground testing with an uncertainty of ~12–14 ns that leads to a range accuracy of <2 m[21]. Further tests in flight enabled to reconstruct a more precise delay time, which was necessary for science operations at Mercury. MESSENGER operated for ~11 years in space, and its instrumentations, including the transponder, coped with the effects of ageing. By interpolating the range data residuals, we were able to determine a linear trend of the time delay that is probably associated with the ageing of the spacecraft transponder. The rate of the mean time delay is ~0.45 ns (~13.5 cm) per year, which provides a maximum offset of <1 m between January 2008 and April 2015. This effect is within the level of accuracy of the range data (1–2 m) that was retrieved during test laboratory results[21].

Another source of range data error is given by station biases due to imperfect calibration. The accurate measurement of the ranging signal round-trip delay is made through digital signal processors at the DSN stations by correlating the uplink and downlink carriers that are coherently related. This calibration may lead to biases on the measured delay with a standard deviation of 1–3 m. To mitigate these calibration errors, range station biases may be estimated in orbit determination. However, the estimation of the range station biases tends to absorb the uncompensated ephemeris mismodeling. For this reason, the range station biases are not estimated in this study of Mercury's ephemeris, heliophysics and GR.

These biases may instead be used to determine the quality of the ephemeris results. After convergence of the global solution, all the adjusted parameters (see Ephemeris and Orbit Determination) are applied in a final iteration, in which the range station biases are adjusted instead of the Mercury's initial state, $GM_\odot$, $J_{2\odot}$, $\beta$, $\eta$, and $G\dot{M}_\odot/GM_\odot$. Figure 3 shows the retrieved station biases that are within the expected range of calibration errors. To compare the quality of these results, Fig. 3 shows the range biases estimated by using JPL DE430 and DE432 original settings.

**Data availability**. The MESSENGER radio tracking data are available from the NASA Planetary Data System archive (http://pds-geosciences.wustl.edu/missions/messenger/rs.htm). The retrieved ephemeris of Mercury is available on the NASA GSFC Planetary Geodynamics Data Archive[30].

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

## Acknowledgements

This work was conducted at the NASA Goddard Space Flight Center. We are grateful to J.B. Nicholas (EST, Inc.), D.E. Pavlis (SGT, Inc.), and D.D. Rowlands (NASA, GSFC) for their help with the GEODYN II software. A.G. thanks L. Iorio (MIUR), and A. Milani, (University of Pisa) for ideas and discussions. The data used in this paper are available at http://pds-geosciences.wustl.edu/missions/messenger/rs.htm.

## Author contributions

A.G., E.M. and S.G. performed radio tracking data processing and preliminary analysis of the MESSENGER orbits. A.G. developed updated relativistic and solar modeling in the NASA GSFC orbit determination software (GEODYN II). A.G., E.M., S.G., F.G.L., G.A.N., D.E.S. and M.T.Z. contributed to the interpretation of the results. A.G. wrote the manuscript with input from E.M., S.G., F.G.L., G.A.N., D.E.S. and M.T.Z.

## Additional information

**Competing interests:** The authors declare no competing financial interests.

