## [Peer Review File · Nature Communications]

Reviewers' comments:

Reviewer #1 (Remarks to the Author):

Review of manuscript "Solar System Expansion and Strong Equivalence Principle as seen by the NASA MESSENGER mission" by Genova et al.

This interesting work is an attempt to use radio tracking data from the mission MESSENGER to Mercury to determine upper limits to the time variation of the Newtonian gravitational constant G (\dot{G} in the text below) and to test the validity of the strong equivalence principle, controlled by the parameter η .

There are two major issues that need to be satisfactorily addressed before publication on Nature Communications can be recommended. Some minor points will be easily tackled.

Major issues.

1) The ability of missions to Mercury to test the strong equivalence principle is still a matter of scientific debate. The authors adopt the approach of Milani et al. (2002), where the position of the solar system center of mass is determined by the gravitational masses. The ensuing indirect acceleration term acting on Mercury and the Earth is the source of a good sensitivity to η .

However, Ashby et al. (2007), with an apparently similar approach, deny the legitimacy of the use of the indirect term, and obtain uncertainties larger by one order of magnitude in the estimate of η . This work is not cited in the manuscript.

The authors should convincingly motivate the reason of their approach, and state why they decided to ignore (Ashby et al., 2007). Alternatively, the authors could quote also a second set of uncertainties when the point of view of Ashby et al. is adopted.

I am reasonably convinced that Milani et al. were correct, but since the issue has not been resolved yet, the alternative approach cannot be ignored in the analysis of real data.

2) At line 521-524 the authors state:

521 ... The
522 orbit of the Earth is not integrated and adjusted in this study since the orbital accuracy of
523 the Earth from the JPL DE432 ephemerides is comparable to the precision of the
524 MESSENGER data.

Assuming that the Earth orbit is perfectly known from solar system ephemerides (DE 430 or DE 432) de facto implies that the location of the solar system barycenter is known. This assumption is very strong and may lead to optimistic uncertainties for \dot{G} and η . The integration of the Earth-Mercury system requires 12 initial conditions. Assuming that six of them are known is not justified. A full simulation with constraints at different levels is mandatory to assess if or by how much the uncertainties quoted in the paper are realistic.

The matter is delicate, because covariance matrices for solar system ephemerides are not available. The authors should run simulations by considering the uncertainties at different levels (I suggest from 3 to 30 m) in order to test the degradation under different assumptions.

Minor issues

67-69: "The SEP states..." SEP and EEP are slightly different. The SEP is an extension of the weak equivalence principle to bodies with not negligible self-gravity. The indistinguishability between gravity fields and accelerated frames is a consequence of this extension. The EEP pertains to non-gravitational experiments in a freely falling lab. There is no need to make reference to the EEP.

70-73 (eq. 2): What is the difference between M_B and m^I ? Isn't M_B the inertial mass? The notation is not clear.

89: Why Jupiter only? There are smaller effects also from other solar system bodies. Perhaps mention that this is the only detectable effect. Or perhaps the authors intended to compare with other three-body systems (Sun-Earth-Moon vs Sun-Mercury-Jupiter).

95-97: "uncorrelated" → "less correlated".

136-137: The determination of the hermean gravity, based on Doppler data, should be largely independent from errors in Mercury's orbit. Doppler data, even if very accurate, are inherently "local", in the sense that they are approximately invariant to small shifts of Mercury's position. The change in velocity would be nearly constant over the time scales of the orbital period. What is the magnitude of the gravity error when the ephemeris of the planet is not estimated?

222: "several orders of magnitude". Only 1.5 to 40.

299-302: Trying with just two solar system ephemerides is not a "sufficient statistics". One could argue that the agreement is just pure chance. The authors should try with other ephemerides, such as INPOP. This is almost a major point that requires convincing evidence.

305-307: "... the discrepancies between the two solutions are slightly larger than the formal uncertainties because of mismodeling of the third body perturbation." It is not at all clear why third body perturbation shall be the culprit. Please clarify.

376: The LT effect is not estimated but just assumed. This implies that GR is assumed true at the 1.5 PN order, which is fine given the sensitivity of MESSENGER. A recent work (Fossat et al., 2017) indicates that the core of the sun is rotating 4x faster than the photosphere. Does this affect the assumed value of the sun's angular momentum to the point that detectable variations are produced in the estimate?

389-390: The cited paper (ref. 32) is wrong. Juno cannot test LT, because Jupiter's pole and its precession is not sufficiently well known. This was pointed out in several conferences.

508: The authors should comment on the weight of flybys in the overall solution. Flybys do not provide good relative positioning between the spacecraft and the planet. Did the authors use just formal covariances for Mercury's state vector? This may lead to optimistic uncertainties for η and \dot{G} if the flybys are driving the global estimate. I do not believe this is the case, but a few words would dissipate legitimate concerns.

568: The term ultra-stable oscillator is generally used in the context of crystal or rubidium clocks. Indicating H-masers explicitly is preferable.

578-582: There is no mention of the time variation of the transponder range bias. What is the effect of ageing on the range bias? Could it possibly affect the estimate?

Reviewer #2 (Remarks to the Author):

The definition of important cosmological and astrophysical parameters relating to the General Theory of Relativity, the gravitational constant and the internal structure of the Sun is considered in this paper. The results are based on new data of the MESSENGER spacecraft near Mercury. The main aim of the paper under review was simultaneously to obtain new, more accurate values of the relativistic parameter β , the Nordtvedt parameter η , the solar quadrupole moment J_2 , as well as the time variation of the Sun gravitational parameter (GM_{\odot}) and to estimate the time variation of the gravitation parameter G from all data of MESSENGER. The authors were not first who obtained estimations of these parameters. However, they obtained these parameters simultaneously from the new MESSENGER data, and with the better accuracy than the previous definitions. Their results show good agreement with the General Theory of Relativity. According to the Mercury motion, the new estimate of the variation of the Solar Gravitational Constant GM has been made, and the value dGM_{\odot}/GM_{\odot} falls within the more narrow interval of values ($1\sigma = 1.41 \times 10^{-14}$ per year). Using previous estimates for changing dM_{\odot}/M_{\odot} of other authors, the new upper bound on the change of the gravitational constant $|dG/G|$ has been obtained, it should not exceed 4×10^{-14} per year.

All these parameters are very significant for cosmology and astrophysics, and will be interesting to others in the wider community.

The paper also includes "Method" with explanations relativistic equations of motion of objects of the Solar system, the Lense-Thirring and Nordtvedt effects, description of MESSENGER data and their processing. All obtained parameters are given with uncertainties derived by the least squares method from observations processing.

The paper is written concisely, with accuracy and clarity.

I recommend this paper for publication.

However, I have several remarks:

1) On line 57 the formula (1) and after line 74 the formulas are not true; "+3" should be changed to "-3".

2) On line 123-126 the phrase is not inaccurate.

Numerical ephemerides EPM (Ephemerides of Planets and the Moon) were originated in 1974, about 10 years later than the DE ephemerides, to support of Russian space flight missions in the Institute of Theoretical Astronomy and then in the Institute of Applied Astronomy of the Russian Academy of Science; and they are being constantly improved at IAA RAS. Please, see .

a) Krasinsky G. A., Aleshkina E. Yu., Pitjeva E. V., Sveshnikov M. L. Relativistic effects from planetary and lunar observations of the XVIII-XX centuries // IAU Symp. N 114, / Relativity in celestial mechanics and astrometry / Eds. Kovalevsky J., Brumberg V. A. Dordrecht: Kluwer Acad. Publ. 1986. P. 315-328.

b) Krasinsky G. A., Pitjeva E. V., Sveshnikov M. L., Chunajeva L. I. The motion of major planets from observations 1769-1988 and some astronomical constants // Celest. Mech. and Dyn. Astron., 1993, V. 55, p. 1-23.

The first paper concerning the IINPOP (IIMCCE) was published only 2008.

The modern EPM ephemerides are:

c) Pitjeva E. V., Pitjev N. P. Relativistic effects and dark matter in the Solar system from observations of planets and spacecraft. - Monthly Notices of the Royal Astronomical Society, 2013, vol.432, issue 4, 3431-3437, DOI: 10.1093/mnras/stt695.

d) Pitjeva E. V., Pitjev N. P. Development of planetary ephemerides EPM and their applications. *Celest. Mech. and Dyn. Astr.*, 2014, 119, 237-256, DOI: 10.1007/s10569-014-9569-0, ISSN 0923-2958.

And see

Hohenkerk C., Arlot J.-E., Kaplan G., Bangert J. A., Bell S. A., Ferrandiz J. M., Fienga A., Folkner W., Lukashova M., Pitjeva E. V., Soma M., Thuillot W., Urban S. E. Historical Reflections on the Work of Commission 4 // *Transactions of the IAU*, 2016, V. 29A,, p. 1-21, DOI: 10.1017/S1743921316000594,

I ask to add on line 124 "...(JPL), Institute of Applied Astronomy of the Russian Academy of Science and the Institut..."

3) On line 192-194

"LLR provided the most accurate determination of β ..."

The better accuracy of β is obtained from all planetary and Mercury ephemerides (your references 19, 25, and our ref. c). The better accuracy of dG/G is obtained from all planetary ephemerides (your references 19, and our ref. c).

4) On Table 1, line after 199

In my opinion, it would be better to add the more accurate estimations:

a) J_2 — ref. 25;

b) β — ref. 25 and c;

c) dG/G — ref c;

5) On line 221 abbreviation ELT appears, and full record only on line 383;

Please add "Einstein – Lense – Thirring" on line 221

6) Online 281-282

", improving the estimates of previous studies by almost order of magnitude"

If to take into account our result from ref. c ($|dG/G| < 8 \times 10^{-14}$ per year), then "improving the estimates of previous studies by two times".

7) Figure 3, Page 14

I don't understand well what are "measurement biases", but they, probably, are about residuals of observations.

In my opinion, your biases for DE430 are too large.

Figure 5 in ref. 17 shows that residuals for DE430 are < 20 m.

It would be well to add that the MESSENGER range data rms is about 0.8 m for modern DE ephemerides (ref. 25). It is about as your result in figure 3.

8) On line 368

Please, add "planets"

"the Sun, Moon, planets, Pluto,)"

9) On line 395

There is any error in the equation (2-M).

Dimensions in the first and second terms of (2-M) are different.

In general, the paper can be recommended for publishing in *Nature Communications* after little correction.

Best regards, Elena Pitjeva.

23.08.2017

Reviewer #3 (Remarks to the Author):

This paper deals with a full orbit reconstruction of the Messenger s/c orbit together with the Mercury orbit in order to evaluate some parameters related to fundamental physics and solar physics.

The authors give a significant improvement of the PPN beta estimation together with $\dot{G}_{\text{Sun}}/G_{\text{Sun}}$.

The paper is well written however some aspects need to be more accurately presented or need further development.

1) Among them, one main issue is coming from the use of equation 1 as an a priori constraint. In doing so, the authors are severely loose in generality.

This means that the obtained results have a sense only for metric theories which represent only a part of the alternative theories of gravitation.

The supplementary Table 1 is very instructive as one can see that the use of eq 1 has a consequence of shifting the 100% correlation between (J_2 and beta) to the 100% correlation between (beta and eta). This is problematic for a least square determination to deal with 100% correlation as the inversion of the normal matrix is then almost impossible. This point should be discussed in the paper somewhere.

Furthermore, most of the other estimations obtained in the literature are obtained without supposing eq 1.

For all these reasons, we strongly suggest to the authors

i) to also give in the Table 2 results obtained without eq1 as a priori constraint but also in considering a fit with J_2 and eta fitted and beta fixed, J_2 and beta fitted and eta fixed. This table should give the values of all the fitted parameters including $\dot{G}_{\text{Sun}}/G_{\text{Sun}}$. In this way, one will see the real uncertainty. It is slightly explained in the text but it will be even clearer if put in the table. This information would be also interesting for comparing the $\dot{G}_{\text{Sun}}/G_{\text{Sun}}$ present estimation with the one published by other teams.

ii) to insist in the text on the limitation of the parameter interpretation induced by the use of eq1.

2) Table 2 should be completed with $\dot{G}_{\text{Sun}}/G_{\text{Sun}}$ obtained with planetary ephemerides such as reference (26) and (19). In these two references, values of $\dot{G}_{\text{Sun}}/G_{\text{Sun}}$ are given but not reported by the authors. In the same manner (19) gives values of J_2 (quite close from the one obtained by the authors) obtained by least squares and stochastic estimation than are not reported in Table 2. The discussions in the text have to be modified accordingly.

About J_2 comparisons, it is also important to stress that the values obtained with planetary ephemerides were obtained without including the ELT. This effect represents about 7% of the J_2 value.

in Table 2, we suppose that there is a misprint in the unit of the last line where one should read 10^{-14} yr and not 10^{13} yr.

4) the idea of changing the initial conditions of the planetary ephemerides to estimate the impact of the perturbations of the other planets on mercury orbit is interesting. However, The authors should try to use DE436 which includes Jupiter and Saturn updates and should be more different from DE432 compared to DE430. Finally by considering two different ephemerides the author address only one side of the problem (perturbations of the planets on Mercury) but they don't consider the impact of changing \dot{G}_{Sun} on the other planet orbits.

general comments:

*) page 9 lines 193. LLR does not provide the most accurate estimation of beta or \dot{G}/G ,

planetary ephemerides do. see (26) or (19).

*) page 12 line 239 and page 22 lines 466 to 471. The fact that it is not possible to disentangle J_2 from β when considering only the Mercury orbit is not due to the limitation of the range accuracy. Correlations are nothing to do with observational accuracies. So maybe the authors should reformulate.

On the other hand, when one estimates PPN parameters, $G\dot{M}_{\text{sun}}/GM_{\text{sun}}$ and J_2 with planetary ephemerides, one disentangles these parameters as we consider different orbits with different semi-major axis and eccentricities. It is obvious when one considers the GR and J_2 effect on the advance of perihelia.

But of course in using only one planet, the correlation is not mild by considering different geometries.

*) page 13 lines 273 to 276. "results were limited to data availability and possible mismodeling of the s/c orbit" the authors have to be more specific and explicative in their critics about these results, that are actually quite consistent with the values presented in this paper.

*) page 14 line 280. "this result strengthens the hypothesis that $G\dot{G}$ is zero". $G\dot{G}$ can be small but nothing clearly demonstrated in this paper that $G\dot{G}$ is zero. It could be small but not zero. the authors have to moderate that statement.

Response to Reviewers

Solar System Expansion and Strong Equivalence Principle as seen by the NASA MESSENGER mission, *Genova et al.*

We would like to thank the three reviewers for their comments and their thorough reading of our manuscript. Here we first highlight the major changes to the manuscript, including extra analysis, and then we give a detailed response to the reviewer comments.

Several of their suggestions required computing a complete new solution, which required ~15 days of wall clock time, using a new Solar System ephemeris in order to demonstrate the solidity of our findings. Reviewer #1 suggested we use other possible planetary ephemerides, as, for example, INPOP17a. However, these ephemerides only account for the orbits and masses of 168 asteroids, whereas the JPL DE ephemerides include 343 asteroids. To test the sensitivity of our solution to different planetary ephemerides, we need to use the same number of celestial bodies in all cases. For this reason, we processed a third case with JPL ephemerides DE436 (as suggested by Reviewer #3) that were originally prepared to improve the definition of the Jupiter system for the Juno mission.

Our additional analysis starting from DE436 confirms our results, leading us to strengthen our conclusions on the Strong Equivalence Principle and the Solar System Expansion. We also further iterated the DE430 case, which enabled us to improve the convergence and retrieval of β and η . The differences between these cases are shown in Table 2, and we also added a supplementary table that reports the solutions with all three JPL DEs.

Reviewer #3 also requested that we show the stability of the solution without the Nordtvedt equation as an *a priori* constraint (that is, without assuming a metric theory of gravitation). We included another supplementary table that shows the stability of the retrieved η and $\dot{M}_{\odot}/M_{\odot}$ values, which are thus independent of the assumption of metric theory of gravitation. However, these results show that the J_2 and β estimates are significantly better when the Nordtvedt equation is applied.

In this document, we will reply (in blue) to the individual comments raised by the reviewers. We also provide a version of the manuscript with changes in red.

Reviewers' comments:

Reviewer #1 (Remarks to the Author):

This interesting work is an attempt to use radio tracking data from the mission MESSENGER to Mercury to determine upper limits to the time variation of the Newtonian gravitational constant G (\dot{G} in the text below) and to test the validity of the strong equivalence principle, controlled by the parameter η .

There are two major issues that needs to be satisfactorily addressed before publication on Nature Communications can be recommended. Some minor points will be easily tackled.

Major issues.

1) The ability of missions to Mercury to test the strong equivalence principle is still a matter of scientific debate. The authors adopt the approach of Milani et al. (2002), where the position of the solar system center of mass is determined by the gravitational masses. The ensuing indirect acceleration term acting on Mercury and the Earth is the source of a good sensitivity to η .

However, Ashby et al. (2007), with an apparently similar approach, deny the legitimacy of the use of the indirect term, and obtain uncertainties larger by one order of magnitude in the estimate of η . This work is not cited in the manuscript.

The authors should convincingly motivate the reason of their approach, and state why they decided to ignore (Ashby et al., 2007). Alternatively, the authors could quote also a second set of uncertainties when the point of view of Ashby et al. is adopted.

I am reasonably convinced that Milani et al. were correct, but since the issue has not been resolved yet, the alternative approach cannot be ignored in the analysis of real data.

We agree with the reviewer that our method is based on the *Milani et al.* approach that includes an indirect perturbation on Mercury's orbit due to η . Our paragraph 1.3 in Methods describes in detail the scientific concept that demonstrates rigorously the η effect on the Solar System Barycenter (SSB). We note that the final formula in that paragraph was not shown explicitly in *Milani et al.* (2002). Actually, the lack of the actual formulation by *Milani et al.* (2002) led, in our opinion, to a scientific debate, which involved Neil Ashby. Ashby published an interesting paper in 2007 comparing his simulation results to Milani's solutions. Paragraph D by *Ashby et al.* (2007) detailed the formulation to determine the range perturbations due to η . Equations 2.21 and 2.22 are consistent with Milani's equations if the correction of SSB is not included. The Appendix of *Milani et al.* (2002) showed, for the first time, the contribution of η to the definition of the SSB, but the formulation was not explained in detail, possibly leading to a misunderstanding. *Ashby et al.* (2007) stated in the text "For η , we have looked for a problem with our results, but have not been able to find one". They do not affirm that Milani's formulation is incorrect or biased, but it seems that they neglected the variation of the SSB due to η .

Milani in 2009 published another paper on the same topic including a detailed formulation of his theory. This study enabled us to reproduce his ideas and have a better understanding on how the SSB is affected by η . We assumed that this debate was resolved after this further publication and this is the reason why we have not included that reference in the text.

Nevertheless, we agree with the reviewer that we should mention *Ashby et al.* (2007) in the Methods section to let the reader know of this alternative study. We think that our paragraph 1.3 strongly motivates our approach showing how the SSB is computed in the

planetary ephemerides. The masses adopted in that calculation are always the gravitational masses, which are retrieved by gravitational effects. The inertial masses should be adopted to determine the barycenter, so a mismatch between gravitational and inertial masses would lead to a correction on the SSB location. We think that this assumption by Milani is valid and supported by the equations in paragraph 1.3. Furthermore, our analysis and interpretation of both formulations allowed us to conclude that the Ashby approach is not complete and, for this reason, it was not used in our investigation.

2) At line 521-524 the authors state:

521 ... The
522 orbit of the Earth is not integrated and adjusted in this study since the orbital
accuracy of
523 the Earth from the JPL DE432 ephemerides is comparable to the precision
of the
524 MESSENGER data.

Assuming that the Earth orbit is perfectly known from solar system ephemerides (DE 430 or DE 432) de facto implies that the location of the solar system barycenter is known. This assumption is very strong and may lead to optimistic uncertainties for \dot{G} and η . The integration of the Earth-Mercury system requires 12 initial conditions. Assuming that six of them are known is not justified. A full simulation with constraints at different levels is mandatory to assess if or by how much the uncertainties quoted in the paper are realistic.

The matter is delicate, because covariance matrices for solar system ephemerides are not available. The authors should run simulations by considering the uncertainties at different levels (I suggest from 3 to 30 m) in order to test the degradation under different assumptions.

The additional analysis with another JPL DE ephemerides in our paper aims also to demonstrate the stability of our solution with significant differences in Earth's orbit.

The Earth ephemeris was not adjusted in our solution, because the Earth orbit is coherently defined in each JPL DE based on a variety of datasets spanning several decades. However, we agree with the reviewer that those ephemerides do not provide uncertainties for orbits, limiting the interpretation on their accuracies.

Our work includes only data from MESSENGER in orbit around Mercury. We do not think that this dataset is sufficient to derive improved knowledge of the Earth orbit compared to the JPL DEs. Furthermore, *Milani et al.* (2002) showed that the 12 components of Earth and Mercury initial conditions lead to a rank deficiency, so only 9 may be adjustable in theory. The assumption that the three additional parameters of the Earth initial state are known from JPL DEs may bias the solution.

For this reason, we tested different ephemerides to assess how changes in Earth position affect $\dot{GM}_{\odot}/GM_{\odot}$ and η . The maximum Earth position differences between DE430 and DE432, and between DE436 and DE432 are ~ 15 m and ~ 80 m, respectively. The figures

below show the evolution of the Earth orbital differences. The solutions with these three different ephemerides provide three solid test cases with larger Earth orbit uncertainties than suggested by the reviewer (3-30m).

The results reported in the Supplementary Table 3 show the stability of β and η estimates, which vary within $1-\sigma$. However, the GM_{\odot}/GM_{\odot} changes with different JPL DEs show that the formal uncertainties for that parameter may be optimistic. It is actually for this reason that we had reported in table 2 (5th column) the effect of a change in solar system ephemerides, which includes different positioning of the Earth.

Minor issues

67-69: "The SEP states..." SEP and EEP are slightly different. The SEP is an extension of the weak equivalence principle to bodies with not negligible self-gravity. The indistinguishability between gravity fields and accelerated frames is a consequence of this extension. The EEP pertains to non-gravitational experiments in a freely falling lab. There is no need to make reference to the EEP.

We removed the reference to EEP as suggested by the reviewer.

70-73 (eq. 2): What is the difference between M_B e m^I ? Isn't M_B the inertial mass? The notation is not clear.

We apologize for the confusion. The parameter has been updated as inertial mass m_i .

89: Why Jupiter only? There are smaller effects also from other solar system bodies. Perhaps mention that this is the only detectable effect. Or perhaps the authors intended to compare with other three-body systems (Sun-Earth-Moon vs Sun-Mercury-Jupiter).

Jupiter causes the larger effect, so it is comparable to the Sun-Earth-Moon case. The other bodies provide a weak perturbation on Mercury's orbit. We slightly rephrased the statement to enhance its clarity.

95-97: "uncorrelated" → "less correlated".

Ok. Modified, thank you.

136-137: The determination of the hermean gravity, based on Doppler data, should be largely independent from errors in Mercury's orbit. Doppler data, even if very accurate, are inherently "local", in the sense that they are approximately invariant to small shifts of Mercury's position. The change in velocity would be nearly constant over the time scales of the orbital period. What is the magnitude of the gravity error when the ephemeris of the planet is not estimated?

The gravity solution is independent from errors in Mercury's orbit if the range data are not included in the analysis. We adopted different JPL DE ephemerides that evolved significantly over the last five years. Discrepancies between DE423, which was used by *Mazarico et al.* (2014), and DE432 are ~100 m in Mercury's position. To mitigate these effects we always estimated range biases that allowed us to compensate for these ephemeris errors. However, we noticed that for several arcs the MESSENGER state rather than the range biases tended to be corrected by tens of meters in order to minimize the range residuals without affecting the Doppler residuals. The offset on Mercury's orbit

was, therefore, projected onto the spacecraft trajectory, leading to errors in the gravity field. Previous gravity solutions that did not include the ephemeris correction showed erroneous features of ~10-20 mGal. This large effect could be related to the sparseness of MESSENGER Doppler data in several arcs, which increases the sensitivity to the range data.

222: "several orders of magnitude". Only 1.5 to 40.

Right. Modified, thank you.

299-302: Trying with just two solar system ephemerides is not a "sufficient statistics". One could argue that the agreement is just pure chance. The authors should try with other ephemerides, such as INPOP. This is almost a major point that requires convincing evidence.

To further demonstrate the stability of our solution, we readjusted the geophysical, relativity and heliophysics parameters using another JPL ephemerides (DE436). This additional case satisfies the comments of both reviewers # 1 and 3. However, we think that the comparison of the results between DE430 and DE432 was already a solid validation since the SSB position in the two ephemerides differs by ~200m. The case with DE436, on the other hand, exhibits a large discrepancy in the Earth's orbit compared to DE430 and DE432 (~80 m) because of the different Jupiter orbit and GM.

We adopted the JPL DEs because they include 343 asteroid masses and orbits. Other planetary ephemerides as, for example, INPOP provide only the trajectory of 168 asteroids (Viswanathan et al. 2017).

305-307: "... the discrepancies between the two solutions are slightly larger than the formal uncertainties because of mismodeling of the third body perturbation." It is not at all clear why third body perturbation shall be the culprit. Please clarify.

We modified the text to emphasize the role of the Earth position and SSB location in the estimation of GM_{\oplus}/GM_{\odot} .

376: The LT effect is not estimated but just assumed. This implies that GR is assumed true at the 1.5 PN order, which is fine given the sensitivity of MESSENGER. A recent work (Fossat et al., 2017) indicates that the core of the sun is rotating 4x faster than the photosphere. Does this affect the assumed value of the sun's angular momentum to the point that detectable variations are produced in the estimate?

The measurement of a larger rotation rate below the convection zone is based on the data of the GOLF instrument onboard SOHO, which have already been used to reveal solar g-mode oscillations (Garcia et al., 2007). However, the feasibility of using GOLF data to detect g modes has been the subject of scientific debate because of the instrument accuracy. Fossat et al. (2017) attempted to develop an alternative approach to estimate

the properties of the core but their results are still questionable. The authors also state: “This rapid rotation nevertheless remains difficult to explain by models describing a pure angular momentum evolution without adding new dynamical processes such as internal magnetic breaking, which could have appeared when the Sun was young (Turck-Chièze et al. 2010).” For this reason, it is difficult to convert this new estimation in a mean rotational rate of the Sun. A rotation of the core 4 times faster than the photosphere does not imply that the rotation of the Sun is 4 times faster. The current spin rotation of the Sun is based on the more accurate observation of p-mode oscillations that also enabled the measurement of the Sun’s gravitational zonal harmonic coefficient J_2 .

However, even if we assume that the angular momentum of the Sun may be 50% larger than the actual knowledge, the accelerations induced on Mercury’s orbit are comparable to accelerations due to J_2 variations of 0.04×10^{-7} , which corresponds to 2- σ as reported in Table 2. The two following figures show indeed the accelerations due to an S_{\odot} increment of $95 \times 10^{39} \text{ kg m}^2 \text{ s}^{-1}$ and a J_2 decrement of 0.04×10^{-7} . The two accelerations are identical in the radial and cross-track directions.

389-390: The cited paper (ref. 32) is wrong. Juno cannot test LT, because Jupiter’s pole and its precession is not sufficiently well known. This was pointed out in several conferences.

We slightly modified the text with an additional reference that shows the difficulties of such an estimate with Juno’s data.

508: The authors should comment on the weight of flybys in the overall solution. Flybys do not provide good relative positioning between the spacecraft and the planet. Did the authors use just formal covariances for Mercury’s state vector? This may lead to optimistic uncertainties for η and G_{dot} if the flybys are driving the global estimate. I do not believe this is the case, but a few words would dissipate legitimate concerns.

We have not applied different weights between flybys and orbital arcs. We understand the reviewer's comment regarding the MESSENGER range data acquired far from the planet. However, the first two flybys were tracked continuously including the closest approach (C/A). The range data collected in proximity of MESSENGER C/As, which were ~200-km altitude (comparable to orbital passes), provide the information necessary for relativistic and heliophysics estimation. The range data during the "wings" of the flyby have already a limited intrinsic weight in our η and GM_{\odot}/GM_{\odot} solution. But those data are important to constrain MESSENGER trajectory during its flyby.

We modified the text to specify that we weighted the range data by accounting for the solar plasma variation during Mercury's orbit with the Sun-Probe-Earth (SPE) angle. This implies that the first flyby is a bit downweighted because the SPE angle was $\sim 30^{\circ}$.

568: The term ultra-stable oscillator is generally used in the context of crystal or rubidium clocks. Indicating H-masers explicitly is preferable.

Modified, thank you.

578-582: There is no mention of the time variation of the transponder range bias. What is the effect of ageing on the range bias? Could it possibly affect the estimate?

To address this comment, we computed a linear interpolation of the range data residuals to determine a possible trend of the transponder time delay due to its ageing. The data show a positive rate of ~ 0.45 ns per year that leads to a time delay differences between January 2008 and April 2015 of ~ 1 m. This value is compatible with test laboratory results (Srinivasan et al., 2007) and it is within the level of accuracy of the range data (1-2m). We added this additional information in Methods.

Reviewer #2 (Remarks to the Author):

The definition of important cosmological and astrophysical parameters relating to the General Theory of Relativity, the gravitational constant and the internal structure of the Sun is considered in this paper. The results are based on new data of the MESSENGER spacecraft near Mercury.

The main aim of the paper under review was simultaneously to obtain new, more accurate values of the relativistic parameter β , the Nordtvedt parameter η , the solar quadrupole moment J_2 , as well as the time variation of the Sun gravitational parameter (GM_{\odot}) and to estimate the time variation of the gravitation parameter G from all data of MESSENGER. The authors were not first who obtained estimations of these parameters. However, they obtained these parameters simultaneously from the new MESSENGER data, and with the better accuracy than the previous definitions. Their results show good agreement with the General Theory of Relativity. According to the Mercury motion, the new estimate of the variation of the Solar Gravitational Constant GM has been made, and the value dGM_{\odot}/GM_{\odot} falls within the more narrow interval of values ($1\sigma =$

1.41×10^{-14} per year). Using previous estimates for changing dM_{\odot}/M_{\odot} of other authors, the new upper bound on the change of the gravitational constant $|dG/G|$ has been obtained, it should not exceed 4×10^{-14} per year.

All these parameters are very significant for cosmology and astrophysics, and will be interesting to others in the wider community.

The paper also includes "Method" with explanations relativistic equations of motion of objects of the Solar system, the Lense-Thirring and Nordtvedt effects, description of MESSENGER data and their processing. All obtained parameters are given with uncertainties derived by the least squares method from observations processing.

The paper is written concisely, with accuracy and clarity.

I recommend this paper for publication.

However, I have several remarks:

1) On line 57 the formula (1) and after line 74 the formulas are not true; "+3" should be changed to "-3".

We apologize for both typos, thank you.

2) On line 123-126 the phrase is not accurate.

Numerical ephemerides EPM (Ephemerides of Planets and the Moon) were originated in 1974, about 10 years later than the DE ephemerides, to support of Russian space flight missions in the Institute of Theoretical Astronomy and then in the Institute of Applied Astronomy of the Russian Academy of Science; and they are being constantly improved at IAA RAS. Please, see .

a) Krasinsky G. A., Aleshkina E. Yu., Pitjeva E. V., Sveshnikov M. L. Relativistic effects from planetary and lunar observations of the XVIII-XX centuries // IAU Symp. N 114, / Relativity in celestial mechanics and astrometry / Eds. Kovalevsky J., Brumberg V. A. Dordrecht: Kluwer Acad. Publ. 1986. P. 315-328.

b) Krasinsky G. A., Pitjeva E. V., Sveshnikov M. L., Chunajeva L. I. The motion of major planets from observations 1769-1988 and some astronomical constants // Celest. Mech. and Dyn. Astron., 1993, V. 55., p. 1-23.

The first paper concerning the IINPOP (IIMCCE) was published only in 2008.

The modern EPM ephemerides are:

c) Pitjeva E. V., Pitjev N. P. Relativistic effects and dark matter in the Solar system from observations of planets and spacecraft. - Monthly Notices of the Royal Astronomical Society, 2013, vol.432, issue 4, 3431-3437, DOI: 10.1093/mnras/stt695.

d) Pitjeva E. V., Pitjev N. P. Development of planetary ephemerides EPM and their applications. Celest.Mech. and Dyn.Astr., 2014, 119, 237-256, DOI: 10.1007/s10569-014-9569-0, ISSN 0923-2958.

And see

Hohenkerk C., Arlot J.-E., Kaplan G., Bangert J. A., Bell S. A., Ferrandiz J. M., Fienga A., Folkner W., Lukashova M., Pitjeva E. V., Soma M., Thuillot W., Urban S. E. Historical Reflections on the Work of Commission 4 // Transactions of the IAU, 2016, V. 29A,, p. 1-21, DOI: 10.1017/S1743921316000594,

I ask to add on line 124 “...(JPL), Institute of Applied Astronomy of the Russian Academy of Science and the Institut...”

We included the Russian Academy of Science in the description of the Ephemeris Groups around the world with an additional reference (*Pitjeva and Pitjev, 2014*).

3) On line 192-194

“LLR provided the most accurate determination of β ...”

The better accuracy of β is obtained from all planetary and Mercury ephemerides (your references 19, 25, and our ref. c). The better accuracy of dG/G is obtained from all planetary ephemerides (your references 19, and our ref. c).

We modified the text to show the improved accuracy on β and $\dot{GM}_{\odot}/GM_{\odot}$ with the latest ephemeris works. Regarding ref. c, we were not aware of those results and their consistency with our findings are definitely relevant to the manuscript. For this reason we included that estimation of \dot{G}/G in Table 1.

4) On Table 1, line after 199

In my opinion, it would be better to add the more accurate estimations:

a) J_2 — ref.25;

b) β — ref.25 and c;

c) dG/G —ref c;

Table 1 shows the various investigations that enable the measurement of those heliophysics and relativistic parameters. The best estimates of those parameters are reported in the main text, which has been modified following the previous comment. We prefer the values of GM , J_2 and β in Table 1 to be compatible with each other however, so we preferred to studies that reported all three parameters with uncertainties. The JPL ephemerides reports and *Park et al. (2017)* do not give, unfortunately, an uncertainty for the GM of the Sun.

However, we cited JPL results in the manuscript as our *a priori* assumption for these parameters.

5) On line 221 abbreviation ELT appears, and full record only on line 383;

Please add “Einstein – Lense – Thirring” on line 221

The definition of the acronym ELT was already in the introduction (line 48).

6) Online 281-282

“, improving the estimates of previous studies by almost order of magnitude”

If to take into account our result from ref. c ($|\dot{G}/G| < 8 \times 10^{-14}$ per year), then “improving the estimates of previous studies by two times”.

We modified the text to specify that \dot{G}/G improvement is about one order of magnitude with respect to Lunar Laser Ranging investigations.

7) Figure 3, Page14

I don't understand well what are “measurement biases”, but they, probably, are about residuals of observations.

In my opinion. your biases for DE430 are too large.

Figure 5 in ref.17 shows that residuals for DE430 are < 20 m.

It would be well to add that the MESSENGER range data rms is about 0.8 m for modern DE ephemerides (ref.25). It is about as your result in figure 3.

The measurement biases are usually estimated during orbit determination to mitigate possible time delays between tracking passes, due to station biases, etc. As explained in the last paragraph in Methods, we used these parameters to determine the quality of the ephemeris results in a final iteration with the adjustment of the range biases instead of Mercury's ephemeris. In all the three cases shown in Fig.3 the range residuals converge to the same level of noise, but in the DE430 post-fit processing the biases have a periodic trend with a maximum amplitude of ~ 60 m.

Folkner et al. (2014) showed the post-fit range residuals in Fig. 5. These range residuals, however, were computed with fixed MESSENGER orbits, which were already converged and fixed. Mismodeling in these pre-converged orbits may have caused these large errors in DE430. When we attempted to use the DE430 to estimate Mercury's gravity field, we needed to adjust range biases larger than 20 m to include the range data in our solution. This is another evidence that a double-integration of planet and spacecraft orbits represents the appropriate technique to estimate Mercury's ephemeris.

8) On line 368

Please, add “planets”

“the Sun, Moon, planets, Pluto,)”

Modified, thank you.

9) On line 395

There is any error in the equation (2-M).

Dimensions in the first and second terms of (2-M) are different.

The second product was a dot product. Thank you for pointing this out.

In general, the paper can be recommended for publishing in Nature Communications after little correction.

Reviewer #3 (Remarks to the Author):

This paper deals with a full orbit reconstruction of the Messenger s/c orbit together with the Mercury orbit in order to evaluate some parameters related to fundamental physics and solar physics.

The authors give a significant improvement of the PPN beta estimation together with $\dot{GM}_{\text{sun}}/GM_{\text{sun}}$.

The paper is well written however some aspects need to be more accurately presented or need further development.

1) Among them, one main issue is coming from the use of equation 1 as an a priori constraint. In doing so, the authors severely loose in generality.

This means that the obtained results have a sense only for metric theories which represent only a part of the alternative theories of gravitation.

The supplementary Table 1 is very instructive as one can see that the use of eq 1 has a consequence of shifting the 100% correlation between (J_2 and beta) to the 100% correlation between (beta and eta). This is problematic for a least square determination to deal with 100% correlation as the inversion of the normal matrix is then almost impossible. This point should be discussed in the paper somewhere.

We modified the text to emphasize that the results in Table 2 are based on metric theories of gravitation. Our study shows that an additional constraint (Nordtvedt equation) is required to enable the estimation of β and J_2 . However, this assumption does not affect the estimation of both η and $\dot{GM}_{\odot}/GM_{\odot}$.

Regarding the correlation between β and η , we do not think that the assumption of the Nordtvedt equation, which forces β to be 0.25 times η , leads to a rank deficiency, but it helps reduce the degrees of freedom of the solution.

Furthermore, most of the other estimations obtained in the literature are obtained without supposing eq 1.

For all these reasons, we strongly suggest to the authors

1. to also give in the Table 2 results obtained without eq1 as a priori constraint but also in considering a fit with J_2 and eta fitted and beta fixed, J_2 and beta fitted and eta fixed. This table should give the values of all the fitted parameters including $\dot{GM}_{\text{sun}}/GM_{\text{sun}}$. In this way, one will see the real uncertainty. It is slightly explained in the text but it will be even clearer if put in the table. This information would be also interesting for comparing the $\dot{GM}_{\text{sun}}/GM_{\text{sun}}$ present estimation with the one published by other teams.

An additional Supplementary Table (ST2) has been added to address the reviewer's comment. This table shows the estimated parameters in all four suggested cases.

2. to insist in the text on the limitation of the parameter interpretation induced by the use of eq1.

We modified the text, stating explicitly that the results in Table 2 are retrieved by assuming a metric theory of gravitation. However, this constraint does not limit the interpretation of the main results of the paper (\dot{GM}_\odot/GM_\odot and η) as demonstrated with the solutions of four different cases in the Supplementary Table 2.

2) Table 2 should be completed with \dot{GM}_\odot/GM_\odot obtained with planetary ephemerides such as reference (26) and (19). In these two references, values of \dot{GM}_\odot/GM_\odot are given but not reported by the authors. In the same manner (19) gives values of J_2 (quite close from the one obtained by the authors) obtained by least squares and stochastic estimation than are not reported in Table 2. The discussions in the text have to be modified accordingly.

We prefer to show only our estimates and reconstructed uncertainties in Table 2. We cited references 19 and 26 in the text, pointing out that those works have provided estimates for \dot{GM}_\odot/GM_\odot with larger uncertainties. Table 1 already shows the J_2 value by *Fienga et al. (2015)*, so we think there is no need to state it again in Table 2.

About J_2 comparisons, it is also important to stress that the values obtained with planetary ephemerides were obtained without including the ELT. This effect represents about 7% of the J_2 value.

in Table 2, we suppose that there is a misprint in the unit of the last line where one should read 10^{-14} yr and not 10^{13} yr.

In Table 2 there was a typo, thank you for pointing this out. We modified the text to address reviewer's comment on the scaling of J_2 values.

4) the idea of changing the initial conditions of the planetary ephemerides to estimate the impact of the perturbations of the other planets on mercury orbit is interesting. However, The authors should try to use DE436 which includes Jupiter and Saturn updates and should be more different from DE432 compared to DE430. Finally by considering two different ephemerides the author address only one side of the problem (perturbations of the planets on Mercury) but they don't consider the impact of changing \dot{GM}_\odot on the other planet orbits.

Following suggestions from reviewers # 1 and 3, we used a third JPL DE to test the stability of our solution. We decided to report the estimates of all three different cases in the new Supplementary Table 3. The \dot{GM}_\odot/GM_\odot estimate changes within ~ 2 -sigma by using the three JPL DEs, confirming that the orbits of the other planets significantly affect the solution. Our study shows that to improve the knowledge of \dot{GM}_\odot/GM_\odot even further ($< 3 \times 10^{-14}$), an accurate modeling of the whole solar system is necessary.

general comments:

*) page 9 lines 193. LLR does not provide the most accurate estimation of beta or \dot{G} , planetary ephemerides do. see (26) or (19).

Modified.

*) page 12 line 239 and page 22 lines 466 to 471. The fact that it is not possible to disentangle J_2 from β when considering only the Mercury orbit is not due to the limitation of the range accuracy. Correlations are nothing to do with observational accuracies. So maybe the authors should reformulate. On the other hand, when one estimates PPN parameters, $GM_{\text{Sun}}/GM_{\text{Sun}}$ and J_2 with planetary ephemerides, one disentangles these parameters as we consider different orbits with different semi-major axis and eccentricities. It is obvious when one considers the GR and J_2 effect on the advance of perihelia. But of course in using only one planet, the correlation is not mild by considering different geometries.

The statement on line 239 does not imply that the accuracy of the range data affects directly the correlation amongst parameters. However, the estimation of η relies on the accuracy of the range data. By applying the Nordtvedt equation, β and η are constrained to have 100% correlation, so the parameter with larger sensitivity (η in this case) drives the other parameter (β) to a lower uncertainty compared to the case with no *a priori* constraint. Range data with higher precision than 1-2 m (as, for example, 20 cm for BepiColombo) would lead to improved formal uncertainty for η (10^{-6} for BepiColombo). A more accurate estimation of η would indirectly enable a reduction in the correlation between β and J_2 . BepiColombo simulations of the relativity experiment showed lower correlation between β and J_2 thanks indirectly to highly accurate range data (Milani *et al.*, 2002). We explained this statement further in Methods at the end of the paragraph on the Strong Equivalence Principle.

We agree with the reviewer that the cause of the correlation between β and J_2 is the fact that we are just using Mercury's ephemeris. The simultaneous determination of different planetary ephemerides would yield a lower correlation.

*) page 13 lines 273 to 276. "results were limited to data availability and possible mismodeling of the s/c orbit" the authors have to be more specific and explicative in their critics about these results, that are actually quite consistent with the values presented in this paper.

We agree with the reviewer that this statement requires a brief explanation. The results presented by Pitjeva and Fienga showed estimates that are consistent with our solution. However, both studies do not include the entire MESSENGER dataset and are based on a "classical" ephemerides solution, which do not co-estimate the orbits of planets and spacecraft. These two distinct advantages of our solution enabled us to improve significantly the GM_{\odot}/GM_{\odot} estimate. The text has been modified.

*) page 14 line 280. "this result strengthens the hypothesis that G_{dot}/G is zero". G_{dot}/G can be small but nothing clearly demonstrated in this paper that G_{dot}/G is zero. It could be small but not zero. the authors have to moderate that statement.

Modified.

Reviewers' comments:

Reviewer #1 (Remarks to the Author):

The authors addressed all issues I raised in my original review. This considerable effort indicates that the quoted uncertainties generally survive the perturbation test, although it would have been desirable to carry out this test using also a non-JPL ephemerides. Although the paper can be published as is in the revised version, I have three remarks that the authors may wish to briefly address in the final publication.

1) It is not clear why the number of bodies must be the same in all ephemerides used as a reference. The formal uncertainties in the PPN parameters should not depend on the ephemerides that are used in the analysis. The same is true for the real uncertainties. If there is a subtlety that requires the use of the same number of bodies (which I do not see), it should be explained.

2) Asby et al. (2007) include the indirect term (see eq. 2.22), although they never call it that way. Note that there is an interesting semi-analytical covariance analysis carried out by Demarchi et al. (2016). It may be worth considering it. (Sorry for not bringing it up earlier.)

3) I have not been able to find Supplementary Table 3 in the revised manuscript. There is a mismatch between the last column in Table 2 for η (6.42) and the corresponding entry in Supplementary Table 3 (6.24).

I congratulate the authors for this interesting paper.

Luciano Iess

Reviewer #3 (Remarks to the Author):

The authors have taken notes of most of the comments proposed by the reviewers. However, they only have partly considered the request related to the choice of planetary ephemerides when more specifically the proposition made by two reviewers to use the INPOP planetary ephemerides with their computation in order to estimate the impact of planetary perturbations on their results. The authors argue that the number of asteroids being different in between the INPOP and DE ephemerides, they choose to not use INPOP ephemerides. This argument does not work as it is exactly what the authors have to test: what is the impact of considering different modeling of the planetary orbits on their results or in saying differently what is the impact of the planetary orbit uncertainties on their results. The DE ephemerides that the authors have considered are very closed from one to another. So their differences do not reflect completely the real uncertainties of planetary orbits. This is the reason why I recommend again strongly to the authors to make their tests in using different ephemerides such as INPOP.

Response to Reviewers

Solar System Expansion and Strong Equivalence Principle as seen by the NASA MESSENGER mission, *Genova et al.*

We would like to thank the reviewers for their additional comments and suggestions. Both reviewers asked for further explanations regarding the number of celestial bodies that are required in our estimation process. Our previous response to Reviewer #1's comment related to the INPOP ephemerides may not have been sufficiently clear since our η and GM_{\odot}/GM_{\odot} solutions do not significantly depend on the additional ~ 150 asteroids masses and orbits that are included in the JPL DEs. However, the comparison of our relativistic and heliophysics results with different planetary ephemerides requires the use of 'test cases' that are self-consistent in the modeling of the solar system. For this reason, we preferred to use JPL DEs that always have the same number of celestial bodies and, furthermore, represent the world standard for the trajectories of planets and asteroids.

As demonstrated in our previous revision, our adoption of the JPL DEs for our processing does not represent an optimistic assumption. Earth and Solar System Barycenter positions discrepancies, for example, between DE430 and DE436 are ~ 80 m and ~ 200 m, respectively. We clearly emphasized these results in order to address the comments, particularly by Reviewer #1. However, we looked further into the differences between the JPL DEs and the INPOP17A ephemerides. By reproducing the plots shown in the technical report of the INPOP17A, we computed the discrepancies of Jupiter's ephemeris (in Right Ascension, Declination, and Geocentric Distance) between DE430 and DE436 (red line), and INPOP17A and DE436 (blue line), respectively. Figure 1 shows larger differences between the JPL DEs than between INPOP17A and DE436. We would like to stress that the mismodeling of Jupiter's orbit represents the main perturbation for the η and GM_{\odot}/GM_{\odot} estimates, and the DE430-436 comparison with the largest Jupiter errors should thus be sufficient to show our independence from the ephemeris used.

Figure 1. Differences in Right Ascension (RA), Declination (DEC) and geodetic distances of the planet Jupiter between DE430 and DE436 (red), and between INPOP17A and DE436 (blue).

To provide additional support for the fact that the JPL DEs significantly differ in their modeling of the solar system, we compared the orbit of Mercury with these same 3 ephemerides. Figure 2 shows that again the different versions of JPL ephemerides show larger differences in Mercury's ephemeris than when comparing INPOP17A to DE436. These larger discrepancies suggest that the differences in the solar system modeling between DE430 and DE436 lead to larger variations in Mercury's orbit with respect to INPOP17A.

In conclusion, we think that an additional solution with INPOP17A is not necessary since the extreme cases are DE430 and DE436, and these more substantial differences sufficiently demonstrate the stability of our relativistic and heliophysics results.

Figure 2. Differences in Right Ascension (RA), Declination (DEC) and geocentric distances of the planet Mercury between DE430 and DE436 (red), and INPOP17A and DE436 (blue).

Below we will reply to the other remarks raised by the reviewers.

Reviewer #1 (Remarks to the Author):

The authors addressed all issues I raised in my original review. This considerable effort indicates that the quoted uncertainties generally survive the perturbation test, although it would have been desirable to carry out this test using also a non-JPL ephemerides. Although the paper can be published as is in the revised version, I have three remarks that the authors may wish to briefly address in the final publication.

1) It is not clear why the number of bodies must be the same in all ephemerides used as a reference. The formal uncertainties in the PPN parameters should not depend on the ephemerides that are used in the analysis. The same is true for the real uncertainties. If there is a subtlety that requires the use of the same number of bodies (which I do not see), it should be explained.

As mentioned above, the inclusion of additional asteroids does not affect our solutions. The uncertainties presented in the manuscript, of course, do not vary with the number of celestial bodies included in our solution. The estimated values may change but always within $1-\sigma$.

2) Asby et al. (2007) include the indirect term (see eq. 2.22), although they never call it that way. Note that there is an interesting semi-analytical covariance analysis carried out by Demarchi et al. (2016). It may be worth considering it. (Sorry for not bringing it up earlier.)

We think that Eq. 2.22 in *Ashby et al. (2007)* partially includes the indirect effect that has been shown by Milani et al. This could be one of the reasons for the discrepancies in their simulation results. We modified the text accordingly. *De Marchi et al. (2016)* is a very interesting paper on the measurement of the Strong Equivalence Principle with the mission BepiColombo. Their results confirm the strong effects of the other planets on the estimation of η . However, the mathematical formulation is still based on *Milani et al.* work so it does not provide additional information on the differences between the approaches by Ashby and Milani.

3) I have not been able to find Supplementary Table 3 in the revised manuscript. There is a mismatch between the last column in Table 2 for η (6.42) and the corresponding entry in Supplementary Table 3 (6.24).

Thank you, there was a typo in Table 2.

Reviewer #3 (Remarks to the Author):

The authors have taken notes of most of the comments proposed by the reviewers. However, they only have partly considered the request related to the choice of planetary ephemerides when more specifically the proposition made by two reviewers to use the INPOP planetary ephemerides with their computation in order to estimate the impact of planetary perturbations on their results. The authors argue that the number of asteroids being different in between the INPOP and DE ephemerides, they choose to not use INPOP ephemerides. This argument does not work as it is exactly what the authors have to test: what is the impact of considering different modeling of the planetary orbits on their results or in saying differently what is the impact of the planetary orbit uncertainties on their results. The DE ephemerides that the authors have considered are very closed from one to another. So their differences do not reflect completely the real uncertainties of planetary orbits. This is the reason why I recommend again strongly to the authors to make their tests in using different ephemerides such as INPOP.

We think that our previous response letter fully addressed the comments raised by the reviewers. Reviewer #3 explicitly suggested to use the JPL DE436 ephemerides as an additional test case. Although this further solution represented a demanding task for our group (~15 days of near-continuous computer time on our cluster to complete), we agreed with both reviewers that an additional case would tighten our conclusions. As demonstrated above, the ephemerides DE430 and DE436 are, in fact, significantly different, and actually show larger discrepancies than INPOP17A. Variations of ~220 m in the Solar System barycenter (as shown between DE430 and DE436), for example, is a very pessimistic condition for the estimation of both η and GM_{\odot}/GM_{\odot} .

The number of asteroids does not affect significantly the solution, as stated above. We agree that our comment may have been misleading. We merely sought consistency in the ephemeris solutions we used.

Below we provide our comments to the reviewer's suggestions that are provided in the manuscript. We used *Italic text* for parts of the manuscript, **red text** for reviewer comments, and blue text for our reply.

“LLR provided accurate estimates of β , η ”

Reviewer #3: “not beta”

Probably the reviewer comment refers to the assumption of metric theory of gravitation by LLR solutions. Although this β solution is based on that hypothesis, the scientific community has extensively used this estimate as reference value for many years.

Table 1 “Latest solution of the INPOP (Intégration Numérique Planétaire de l’Observatoire de Paris) planetary ephemerides”

Reviewer #3: “as the author give \dot{GM}_\odot/GM_\odot , they should also give competitive values in this table like the one from inpop and epn.”

and

Table 1 “Ephemerides of the Planets and the Moon”

Reviewer #3: “why not the one from inpop?”

This table reports the best estimates of each single parameter described in the manuscript. We prefer to include only \dot{G}/G and \dot{M}_\odot/M_\odot .

We included the *Fienga et al. 2015* reference for \dot{G}/G .

“both parameters are equal to 0 (Supplementary Table 2)”

Reviewer #3: “this table should not be put on the supplementary material but is part of the main discussion.”

We think the Supplementary Table 2 should not be part of the main manuscript since it provides only additional cases that confirm the stability of the solution reported in Table 2. Interested readers who want to see the variations between the three solutions are referred to this supplementary table.

Table 2 “ $\dot{GM}_\odot/GM_\odot (x10^{-14} yr^{-1})$ ”

Reviewer #3: “the authors must give the \dot{GM}_\odot/GM_\odot values obtained by other teams here or the table 1.”

We already cited the papers from other groups in the text, so we see no need to repeat those in the tables.

Table 2 “A priori and estimated values, and uncertainties from the global estimation of the GR and..”

Reviewer #3: “no. in the frame of metric theories in assuming the Nordvedt equation”

We added “by assuming a metric theory of gravitation”.

Table 2 “by using the JPL DE430, DE432 or DE436 ephemerides to mode..”

Reviewer #3: “these ephemerides are very similar. The author have to change their ephemerides to some different one, INPOP17a for ex. The fact that this ephemerides does not have the same number of asteroids does not play a role.”

We do not agree that DE430 and DE436 are very similar, as demonstrated above.

“estimates, were limited by the data availability and possible mismodeling of spacecraft orbits. Our..”

Reviewer #3: “not agree. there is no demonstration of this claim s/c mismodeling in the previous

estimations. the authors have to remove this sentence.”

And

“Furthermore, our new technique, which consists of a double integration and a combined estimation of both planet and spacecraft orbits, mitigates the systematic errors related to the spacecraft position and velocity.”

Reviewer #3: “not clearly demonstrated with this paper. In the opposite, it has been shown in Verma 2014, that a separate estimation of s/c orbit followed by a determination of station bias and planetary ephemerides construction was not degraded at the condition that the planet gravity field was updated. So what the author can argue is that they have a better updated gravity field.”

We stated in the paper that the combined estimation of the MESSENGER and Mercury orbits enabled us to enhance not only the relativistic and heliophysics solutions but also the geophysical results, which are the main topic of a separate paper focused on the interior structure of Mercury. Mazarico *et al.* 2014 showed in detail our precise orbit determination approach for the orbit of MESSENGER that allowed us to provide an accurate estimation of Mercury’s gravity field. This paper showed the significant effect of solar and thermal radiation pressure that was mitigated by adjusting the areas of the spacecraft panels. In paragraph 2 in Methods of our manuscript, we provide information on our MESSENGER orbit determination that is similar to the Mazarico *et al.* 2014 approach for the spacecraft trajectory.

Verma *et al.* 2014 presented an interesting paper on the ephemeris of the planet Mercury. This paper was based on the first years of the MESSENGER mission, so before the low-altitude campaign when the spacecraft was tracked at altitudes as low as ~25 km, with stronger gravitational effects. Furthermore, Figure 2 of Verma *et al.* 2014 shows very high solar radiation scale factors (>2, although the nominal value should be 1). These large errors that we cited in the manuscript are indeed related to the spacecraft macro-model. Uncompensated radiation effects may significantly affect the orbit of the spacecraft and, consequently, the reconstructed ephemeris. We may not have provided an explicit demonstration for the spacecraft mismodeling effects, but we think that the information included in the manuscript, especially when compared to earlier results, is enough to support our thesis (also note we said “*possible mismodeling of spacecraft orbits*”).

“The discrepancy between our solution and the computed $\dot{M}_{\odot}/M_{\odot}$ may be interpreted as an indirect measurement of the universal constant time-variation.”

Reviewer #3: “yes but there is also a part induced by the correlation between parameters and by uncertainties induced by the earth fixed and other planet fixed orbits.”

Agreed, but these caveats are already described in the manuscript and methods.

“This result strengthens the hypothesis that \dot{G}/G is close to 0, improving the estimates of LLR studies by almost an order of magnitude”.

Reviewer #3: “this should be partly true if the authors make the same run with different planetary ephemerides.”

As above, the three cases provide a good basis for estimating the levels of uncertainty in our solutions.

“using DE430 or DE436 for”

Reviewer #3: “again these ephemerides are very close. The authors have to use a different model such as inpop17a, including different asteroid masses determination.”

We used the DE436 because Reviewer #3 previously suggested: *“The authors should try to use DE436 which includes Jupiter and Saturn updates and should be more different from DE432 compared to DE430. Finally, by considering two different ephemerides the author address only one side of the problem (perturbations of the planets on Mercury) but they don't consider the impact of changing \dot{G}_{Sun} dot on the other planet orbits.”* By providing this additional third

case we addressed this comment and significantly improved the robustness and interpretation of our results. As stated above using Figure 1 and 2 in this reply, the DE430 and DE436 ephemerides are not as close as perhaps expected.

“enhances the knowledge of the relativistic parameters β and η ”

Reviewer #3: “not beta, because of the Nordvedt equation.”

We removed β .

“The negative rate of $\dot{GM}_{\odot}/GM_{\odot}$ is very close to theoretical computations of the Sun’s mass loss rate leading us to significantly constrain the universal constant time-variation.”

Reviewer #3: “not really. This says that the \dot{G}/G is small.”

Inferring that \dot{G}/G is smaller than 4×10^{-14} is an important constraint.

Reviewers' comments:

Reviewer #1 (Remarks to the Author):

I have no objections to the publication of the paper in its last version.

The work would have been definitely stronger if a different number of bodies were considered in the planetary ephemerides, and those ephemerides were generated using different codes.

Selecting the papers to be cited is a free choice of the authors. I just point out that the abstract of Marchi et al. (2016), published on Phys Rev. D., seems quite relevant to the discussion:

... "The aim of the relativity experiment is the measurement of the post-Newtonian parameters. Thanks to accurate tracking between Earth and spacecraft, the results are expected to be very precise. However, the outcomes of the experiment strictly depend on our "knowledge" about solar system: ephemerides; number of bodies (planets, satellites, and asteroids); and their masses. In this paper we describe a semianalytic model used to perform a covariance analysis to quantify the effects on the relativity experiment, due to the uncertainties of Solar System bodies' parameters. In particular, our attention is focused on the Nordtvedt parameter η used to parametrize the strong equivalence principle violation." ...

Reviewer #3 (Remarks to the Author):

Auhtors have taken care of some of the propositions that were made. A work of comparisons between ephemerides was made but missing the point of the asteroid modeling. As the scope of this paper is not planetary ephemerides, this approach can be accepted but as to be mentionned in some manner in the text with a sentence mentionning these comparisons.

However, for other aspects, the authors did not take into account some of propositions of modifications or moderations that were proposed.

Few of them are recalled here and have to be made before publication:

1) "LLR provided accurate estimates of β , η "

Reviewer #3: "not beta"

Probably the reviewer comment refers to the assumption of metric theory of gravitation by LLR solutions. Although this β solution is based on that hypothesis, the scientific community has extensively used this estimate as reference value for many years.

This is not a good argument for continuing the wrong assumption that LLR provides accurate estimates of beta. It must be replaced.

3)"both parameters are equal to 0 (Supplementary Table 2)"

Reviewer #3: "this table should not be put on the supplementary material but is part of the main discussion."

We think the Supplementary Table 2 should not be part of the main manuscript since it provides only additional cases that confirm the stability of the solution reported in Table 2. Interested readers who want to see the variations between the three solutions are referred to this supplementary table.

The table 2 is not just a collection of supplementary cases tested by the authors. Table 2 provides more general results than those proposed by the authors in their main part of the manuscript. This Table 2 must be put into the main section of the paper.

3) Table 2 "by using the JPL DE430, DE432 or DE436 ephemerides to mode.."

Reviewer #3: "these ephemerides are very similar. The author have to change their ephemerides to some different one, INPOP17a for ex. The fact that this ephemerides does not have the same number of asteroids doe not play a role."

4)Reviewer #3: "yes but there is also a part induced by the correlation between parameters and by uncertainties induced by the earth fixed and other planet fixed orbits."

Agreed, but these caveats are already described in the manuscript and methods.

Yes but the previous statement has to be moderated in recalling these questions related to correlations

We do not agree that DE430 and DE436 are very similar, as demonstrated above.

They are similar on the main important aspect here : the modeling of asteroid perturbations. A sentence stressing that point should be added in the text.

Response to Reviewers

Solar System Expansion and Strong Equivalence Principle as seen by the NASA MESSENGER mission, *Genova et al.*

We would like to thank the reviewers for their thorough reading of the manuscript and their comments. The attached manuscript includes changes that address the latest suggestions of the reviewers. Table 2 was not modified because the results reported in the Supplementary Table 2 readily shows the influence of our assumptions on β and J_2 of the Sun, which is not the main topic of the paper. Table 2 is focused on the main findings of our own work, which are the measurement of the solar system expansion due to the mass loss rate and the Strong Equivalence Principle. The Supplementary Table 2 is referred to from the main manuscript for interested readers who want to have a better understanding of the stability of our results.